# Structural basis for UFM1 transfer from UBA5 to UFC1

Manoj Kumar[1], Prasanth Padala[1,6], Jamal Fahoum[1], Fouad Hassouna[1], Tomer Tsaban[2], Guy Zoltsman[3], Sayanika Banerjee[1], Einav Cohen-Kfir[1], Moshe Dessau [4], Rina Rosenzweig [3], Michail N. Isupov[5], Ora Schueler-Furman [2] & Reuven Wiener [1✉]

Ufmylation is a post-translational modification essential for regulating key cellular processes. A three-enzyme cascade involving E1, E2 and E3 is required for UFM1 attachment to target proteins. How UBA5 (E1) and UFC1 (E2) cooperatively activate and transfer UFM1 is still unclear. Here, we present the crystal structure of UFC1 bound to the C-terminus of UBA5, revealing how UBA5 interacts with UFC1 via a short linear sequence, not observed in other E1-E2 complexes. We find that UBA5 has a region outside the adenylation domain that is dispensable for UFC1 binding but critical for UFM1 transfer. This region moves next to UFC1's active site Cys and compensates for a missing loop in UFC1, which exists in other E2s and is needed for the transfer. Overall, our findings advance the understanding of UFM1's conjugation machinery and may serve as a basis for the development of ufmylation inhibitors.

[1] Department of Biochemistry and Molecular Biology, The Institute for Medical Research Israel-Canada, Hebrew University-Hadassah Medical School, Jerusalem 91120, Israel. [2] Department of Microbiology and Molecular Genetics, Institute for Medical Research Israel-Canada, Faculty of Medicine, The Hebrew University of Jerusalem, Jerusalem 91120, Israel. [3] Department of Chemical and Structural Biology, Weizmann Institute of Sciences, Rehovot, Israel. [4] Azrieli Faculty of Medicine, Bar-Ilan University, Safed 1311502, Israel. [5] The Henry Wellcome Building for Biocatalysis, Biosciences, University of Exeter, Stocker Road, Exeter EX4 4QD, United Kingdom. [6] Present address: Department of Biochemistry, School of Biomedical Sciences, University of Otago, Dunedin 9016, New Zealand. ✉email: reuvenw@ekmd.huji.ac.il

Post-translational modifications by ubiquitin-like proteins (UBLs) are essential regulatory mechanisms in eukaryotes[1–4]. Ubiquitin fold modifier 1 (UFM1) is a UBL that shares as little as 21% sequence identity with ubiquitin, but still possesses ubiquitin's classic β-grasp fold[5,6]. Modification by UFM1 (ufmylation) is involved in regulating several cellular processes such as the DNA damage response, the endoplasmic reticulum stress response, cell division, erythropoiesis, and fatty acid metabolism[7–12]. Impaired ufmylation leads to a range of cellular dysfunctions and has been shown to be connected to several human diseases including cancer and diabetes[13–16]. Analogous to ubiquitination, ufmylation is carried out by a three-enzyme cascade involving E1, E2, and E3 proteins. The E1 UBA5 has three distinct regions: (1) an adenylation domain that binds ATP and magnesium ions; (2) a UFM1-interacting sequence (UIS) spanning amino acids 334 to 346 that holds UFM1 in a trans binding mechanism; and (3) a UFC1-binding sequence (UBS) from 392 to 404 that binds UFC1[17–21] (Fig. 1a, b). The process of UFM1 activation is initiated by the binding of ATP, magnesium ions, and UFM1 to UBA5, followed by adenylation of the C-terminal glycine of UFM1 with the release of pyrophosphate. The adenylated UFM1 is then subjected to a nucleophilic attack by the catalytic C250 of UBA5 to form a thioester bond with the C-terminal glycine of UFM1. This is followed by a trans thiolation process, whereby the UFM1 is transferred to the E2 UFC1 forming a thioester bond with C116 of UFC1. The final stage includes the transfer of UFM1 to the target protein via the E3 ligase UFL1[22].

The mechanism of ufmylation is unique in many ways, one of which is how UFC1 binds UBA5 in order to promote UFM1 transfer. Canonical E1 enzymes contain a ubiquitin fold domain (UFD) that is dedicated to the interaction with the E2[23,24]. Similarly, in the non-canonical E1 enzyme Atg7, a specific domain (~280 amino acids), known as the N-terminal domain (NTD), is responsible for the interactions with the cognate E2s[25–27]. However, UBA5 does not harbor such a domain but instead has a short sequence called UBS that is responsible for the interaction with UFC1[28]. To date, there have been no structural insights into how this short sequence binds UFC1 and ensures UFM1 transfer specificity.

Here we report the crystal structure of UFC1 in complex with the UBS of UBA5. This structure reveals that the UBS adopts a helical conformation that interacts with UFC1, mainly via hydrophobic interactions. UFC1 possesses a pocket, lacking in other E2 enzymes, in which the UBS fits spatially. Moreover, we have identified in UBA5 a region outside the UBS that is paramount for the transfer of UFM1. Upon UBS binding to UFC1, this region approaches the catalytic C116 and ameliorates the

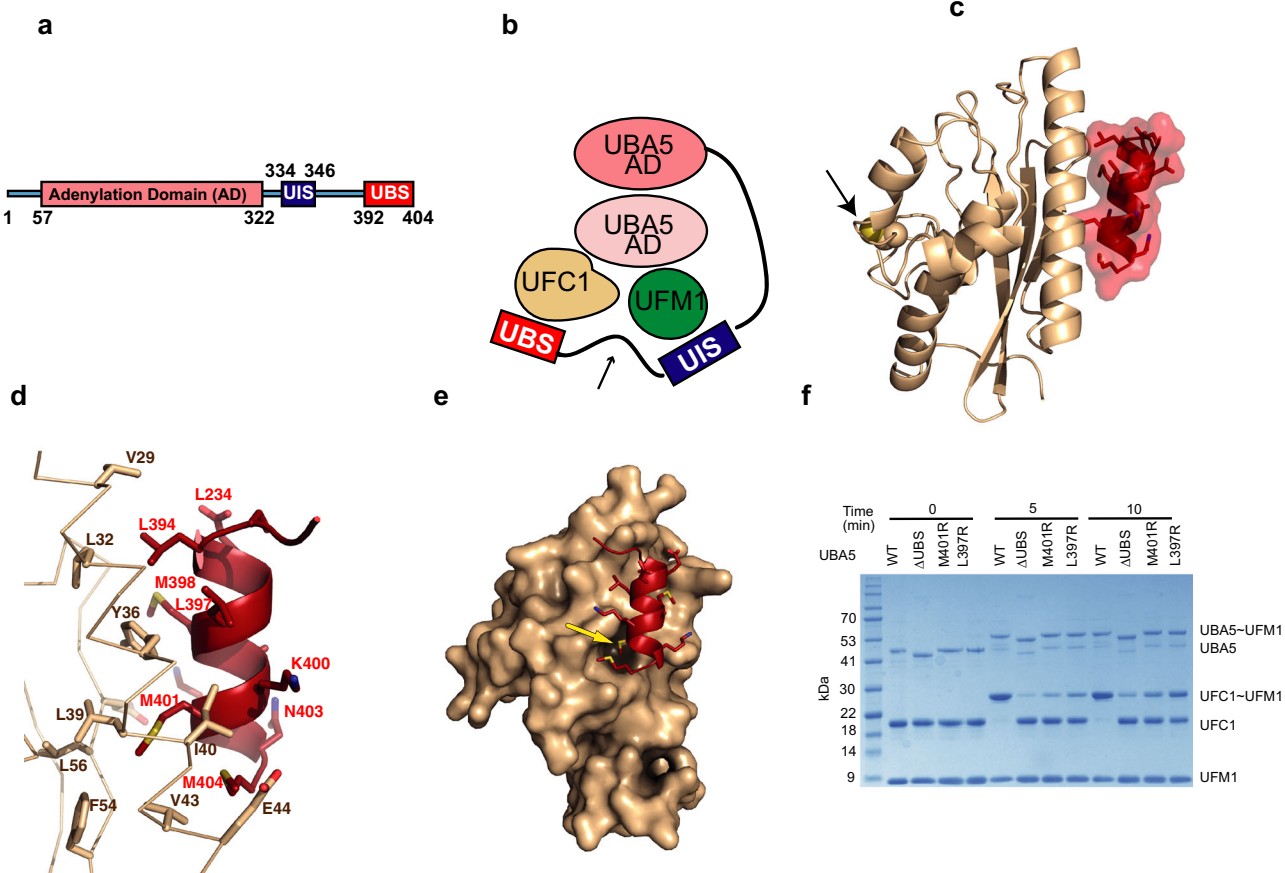

**Fig. 1 Structural characterization of the UFC1-UBS interaction. a** Schematic representation of UBA5 elements. UIS (UFM1-interacting motif); UBS (UFC1-binding sequence) **b** Trans-binding mechanism of UFM1 and UFC1 to the homodimeric UBA5. UFM1 and UFC1 bind the UIS and UBS, respectively, of one molecule of UBA5 and interacts with the adenylation domain of the other UBA5 molecule. For simplicity, the UIS and UBS are shown only for one molecule of UBA5. The black arrow indicates the linker connecting the UIS to UBS; AD-adenylation domain. **c** Crystal structure of UFC1 (brown) bound to UBS (red); black arrow points on the active site Cys116 of UFC1 (yellow sphere). **d** Contact between UBS and UFC1 α-helix 1 and β-strand 1. Sidechains of UFC1 residues involved in UBS binding are shown in stick representation. **e** M401 (highlighted by yellow arrow) of UBS (red) occupies a hydrophobic pocket on the surface of UFC1 (brown) **f** SDS-PAGE analysis showing the effect of UBS mutations on UFM1 transfer to UFC1. The gel is representative of two independent experiments.

physicochemical features of that cysteine. We found that this region is mandatory to enable the UFC1 active site Cys to attack and accept UFM1. Surprisingly, UFC1 lacks an essential loop located next to the active site Cys that exists in other E2 enzymes. Our results suggest that UBA5 contributes to this region to compensate for the missing loop in UFC1, thereby permitting UFM1 transfer. Overall, we have deciphered a mechanism that guarantees UBL transfer specificity between E1 and E2 that is based on an E2 that lacks a key element needed for the catalytic activity that is complemented by its cognate E1.

## Results

**Crystal structure of the UFC1-UBS complex.** To provide structural insights into the interaction of UFC1 with the UFC1-binding sequence (UBS) of UBA5, we determined the crystal structure of UBA5 (389-404) fused to the C-terminus of UFC1 at 2.4 Å resolution (Supplementary Table 1). The asymmetric unit comprises two molecules of UFC1, each interacting with a UBS that is fused to a UFC1 molecule derived from another asymmetric unit (Supplementary Figs. 1 and 2). The UBS forms an α-helix and interacts with UFC1 on the side opposite to the active site surface (Fig. 1c). The UBS residues L394, L397, M398, M401, and M404 form hydrophobic interactions with residues V29, L32, L39, I40, V43 of α-helix 1 and F54 and L56 of β-strand 1 of UFC1 (Fig. 1d). More specifically, M401 occupies a hydrophobic pocket on the UFC1 surface (Fig. 1e). Accordingly, M401R and L397R mutations in the UBS prevented the interaction with UFC1 (Supplementary Fig. 3) and damaged UFM1 transfer (Fig. 1f). Besides hydrophobic interactions, the main chain carboxyl oxygen atoms of M404 forms hydrogen bonds with the side chains of UFC1 K47 and R55. In addition, K400 and K402 of the UBS form salt bridges with E44 and E57 of UFC1, respectively. However, mutations of these residues did not affect UFM1 transfer (Supplementary Fig. 4), suggesting that the binding of UBS to UFC1 is mainly mediated by hydrophobic interactions.

To date, UBA5 has been shown to transfer UFM1 to UFC1 but not to other E2 enzymes[6]. Accordingly, the binding surface that is responsible for the interaction with the UBS does not exist in other E2 enzymes, suggesting that they are not amenable for UBS binding. Specifically, the superposition of UFC1-UBS fused structure with other E2s shows that the UBS suffers steric clashes with α-helix 1, β-strand 1, and the loop connecting the strands-1 and 2 (Supplementary Fig. 5).

**A UBS-independent role of the UBA5 linker.** While the UBA5 construct 1–392, which lacks the UBS, significantly hampered the transfer of UFM1, we unexpectedly found that UBA5(1–347), which lacks not only the UBS but also the linker connecting the UBS to the UIS, completely failed to transfer UFM1 to UFC1 (Fig. 2a). Moreover, removing this linker, but not the UBS (UBA5Δ349–389) abolished transfer of UFM1 (Fig. 2a) but had little effect on the binding of UFC1 to UBA5 (the measured Kd was 5.5 μM while the one we published for the WT UBA5-UFC1 complex was 1 μM)[29] (Fig. 2b). These results suggest that UBA5 amino acids 349–389 are not essential for UFC1 binding, but have an additional role besides serving as a linker that connects the UBS. Therefore, to further characterize the UBS-independent role of amino acids 347–392 in UFM1 transfer, we tested UBA5 truncations for their ability to transfer UFM1. As shown in Fig. 2a, while UBA5(1–363) failed to transfer, UBA5(1–377) functions similarly to UBA5 (1–392). These results suggest that residues 363–377 are indispensable for UFC1 transfer. Next, we asked whether UBA5 residues 347–377 play a role in UFM1 transfer to another acceptor besides UFC1. Since to date, UFC1 is the only known E2 that works with UFM1, we exploited free Cys

as a UFM1 acceptor. As shown in Fig. 2c, free Cys at a high concentration successfully discharged UBA5 (1–347), suggesting that residues 347–377 are critical for UFM1 transfer to UFC1 but not to free Cys.

Next, with these results in hand, we hypothesized that residues 347–377 are required not for UBA5 but for UFC1 activity. Specifically, if these residues are needed for UFC1 function, then one would expect that adding them in trans would recover transfer to some extent. Indeed, as shown in Fig. 2d, e, we recovered the transfer of UFM1 from UBA5(1–347) to UFC1 by adding a UBA5 fragment comprising residues 347–404. Of note, since UBA5 fragment 347–404 does not run in SDS–PAGE analysis as expected for a 6.4 kDa protein (Fig. 2d), we confirmed its molecular weight in solution using SEC-MALS analysis (Supplementary Fig. 6). Taken together our results suggest that residues 347–377 that are critical for transfer can satisfy their role even without being covalently linked to the UBA5 protein.

**The UFC1 active site Cys is highly solvated.** To elucidate how residues 347−377 of UBA5 contribute to UFC1 activity, we asked whether UFC1 possesses any unique feature/s that are not present in other E2 enzymes. Initially, we calculated the buried surface area of the E2 active site Cys (Supplementary Table 3). Surprisingly, we found that C116 is highly exposed to the solvent compared to the relatively buried catalytic cysteine in other E2 proteins. Specifically, the buried surface area of UFC1 C116 is 4% while for other E2s this value is significantly higher (15–36%). Similar to other E2s, UFC1 possesses a ubiquitin-conjugating (UBC) domain[30,31]. However, UFC1 contains an additional N-terminal helix (α0) which has been shown to increase protein stability but does not affect UFM1 transfer[32]. Following the active site Cys, UFC1 has an α-helix (α1-2) while in other E2s there is a $3_{10}$ helix (Fig. 3a). In addition, UFC1 lacks two helices at the C-terminus (α3 and α4). α-helix 2 (α2) of UFC1 is long and unbroken, while in other E2s this helix is shorter (Fig. 3b, c and Supplementary Figs. 7 and 8). In these E2s, α2 continues with a loop that links to helix 3 (α3). This loop envelops the catalytic cysteine, thus burying the residue in other E2s. Accordingly, as expected for highly exposed Cys, we found that the predicted pKa of UFC1 C116 is 8.5, a value equivalent to a free cysteine, and ~1.5 pKa units lower than all other E2s (Supplementary Table 3). Overall, our results suggest that UFC1 possesses a highly exposed active site Cys that has a lower pKa than the other E2s.

**The UBA5 linker plays a role in UFC1 active site desolvation.** The lack of a loop next to the UFC1 active site makes the latter heavily solvated by the electrophilic protons supplied by the water. This causes the active site Cys to be less potent for the nucleophilic attack, and thereby requires desolvation prior to its nucleophilic attack[33–36]. This motivated us to investigate whether UBA5 residues 347–377 play a role in UFC1 active site desolvation. Initially, we asked whether we can overcome the defects in UFM1 transfer from UBA5(1–363) by increasing the pH. Here the idea is that elevating the pH will make the UFC1 active site Cys a better nucleophile (the ratio of thiolate($S^−$) to thiol (SH) increases). This, in turn, will allow a productive nucleophilic attack even though the active Cys is highly solvated. Indeed, as shown in Fig. 3d, while UBA5 1–363 failed to transfer UFM1 to UFC1 at pH 6.5, at pH 7.5, and above transfer was detected. Of note, we ascribe the ability of free Cys to discharge UBA5 at pH 6.5 (Fig. 2c), to its high concentration (100 mM), which increases the availability of unsolvated free thiol to perform the nucleophilic attack. Accordingly, when we performed the transfer assay with 500 μM UFC1, a 100-fold increase from the regular concentration, a minor transfer of UFM1 from UBA5(1−363) to

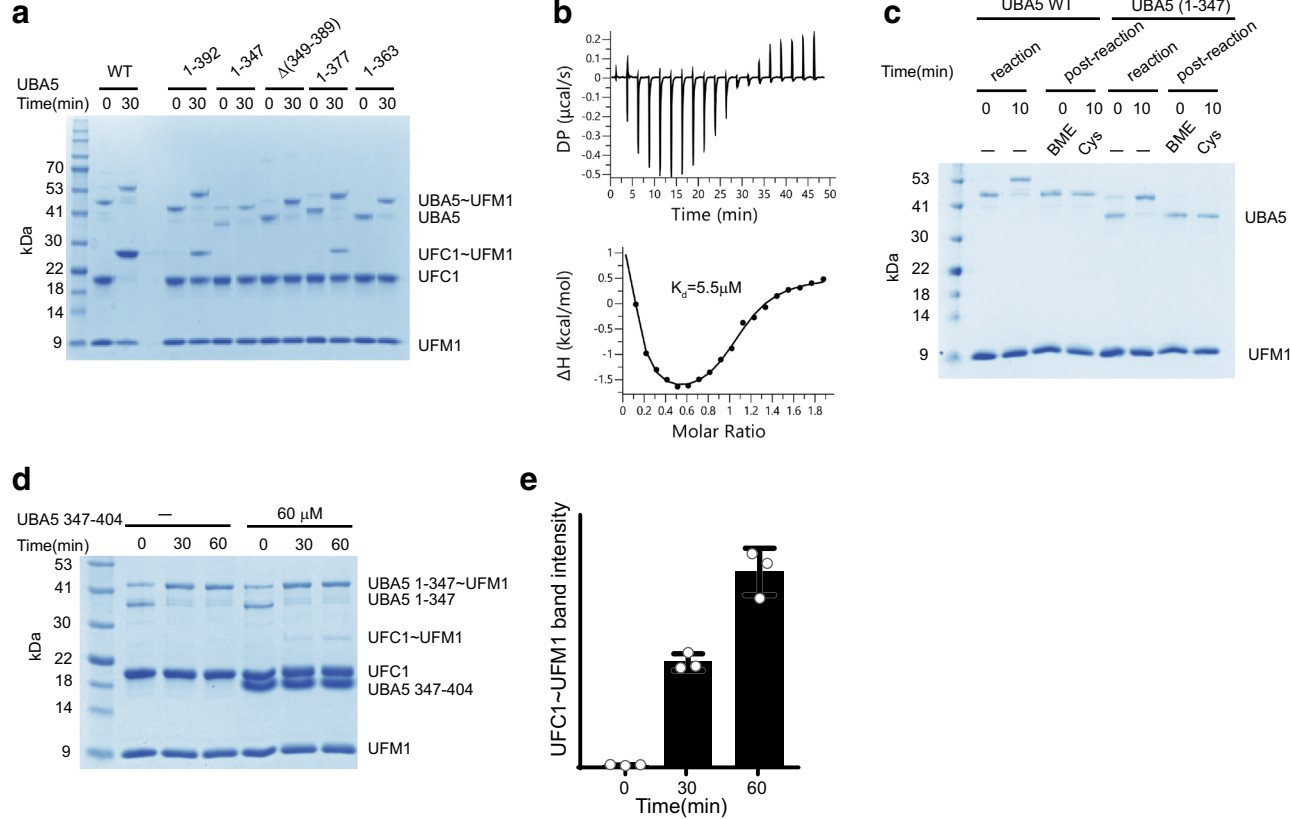

**Fig. 2 The UBS independent role of the linker connecting the UBS to UIS. a** SDS–PAGE analysis showing the effect of UBA5 truncations or deletion on the transfer of UFM1 to UFC1. The gel is representative of at least two independent experiments. **b** ITC experiment of UFC1 binding to UBA5 Δ349−389. The top graph represents raw data of heat flow versus time. The area under the peaks of the upper panel was integrated and plotted as kcal per mole of UFC1 as a function of binding stoichiometry in the bottom panel. Thermodynamic parameters are summarized in Supplementary Table 2. **c** SDS–PAGE analysis showing the discharge of UBA5 WT or (1–347) by free cysteine (100 mM). The gel is representative of at least two independent experiments. **d** SDS-PAGE analysis showing the effect of UBA5(347–404) on the transfer of UFM1 to UFC1 from UBA5(1–347). **e** Three independent experiments as shown in **d** were performed to quantify the effect of adding the UBA5 fragment in trans on UFM1 transfer to UFC1. Data are represented by mean ± SD.

UFC1 was observed. Taken together, our results support that UBA5 residues 363–377 play a role in UFC1 active site Cys desolvation.

In order to fulfill its role in UFC1 desolvation, UBA5 residues 347–377 have to reach the vicinity of the UFC1 active site. Therefore, to gain structural insights about this region and how it interacts with UFC1, we attempted to crystalize UFC1 in complex with UBA5 347–404, but with no success. However, we successfully crystallized and determined the structure of UBA5(347–404) that is fused to the N-terminus of UFC1 to 2.65 Å resolution (Supplementary Table 1). The asymmetric unit contains one fusion molecule, and UFC1 interacts with the UBS that arrives from another asymmetric unit (Fig. 3e). As expected, the interaction of the UBS with UFC1 retains the same architecture as observed in the structure with the shorter fragment containing only the UBS sequence (Fig. 1). In both structures, the UBS binds to the α-1 helix and β-1 strand of UFC1. However, for the linker connecting the UBS, we obtained electron density for residues 382–389 but not for 347–381, suggesting that most of the linker is mobile. V382 (the first built residue at the N-terminus of UBA5) is 35 Å away from UFC1 C116 (Fig. 3e). While this distance is far from the active site, UBA5 residues 381–363 can span it, thereby we cannot rule out the possibility that the linker reaches the vicinity of UFC1 active site Cys.

To study how the UBA5 linker plays a role in UFC1 desolvation, we analyzed the conservation score of its residues.

Using Conseq[37] we found that residues G367-T373 are relatively more conserved (Fig. 4a). Moreover, a biallelic mutation A371T has been reported to have negative implications on ufmylation in human brain development, leading to early onset of encephalopathy in infants, further supporting the importance of that region[38]. Therefore, to start understanding the role of this region we mutated the conserved Y372. While Y372F showed transfer comparable to wild type, Y372A or Y372E had severely decreased ability to transfer (Fig. 4b, c), but as expected no effect on UFC1 binding (Fig. 4d). This suggests that the aromatic residue at position 372 of UBA5 is critical for this region to be able to desolvate the UFC1 active site.

To satisfy its role in desolvation, Y372 has to reach the active site Cys; however, our binding experiments show that residues 349–389 are dispensable for UFC1 binding (Fig. 2b), suggesting that the interaction of UBA5 Y372 with UFC1 is transient. To overcome the transience of this interaction, we investigated whether we could trap it using a disulfide bond. Since the UBA5 active site Cys should be juxtaposed with the UFC1 active site Cys for transthiolation, we successfully obtained a disulfide bond between UBA5 WT and UFC1 active site Cys residues (we used DPDS to facilitate disulfide bond formation; see Method section) (Fig. 4e). Accordingly, UBA5 with a C250A mutation, although comprising 7 cysteine residues in its sequence, failed to form a disulfide bond with UFC1. However, the double mutant C250A and A371C recovered the disulfide bond with UFC1, suggesting that amino acid 371 of UBA5 reaches the vicinity of the UFC1

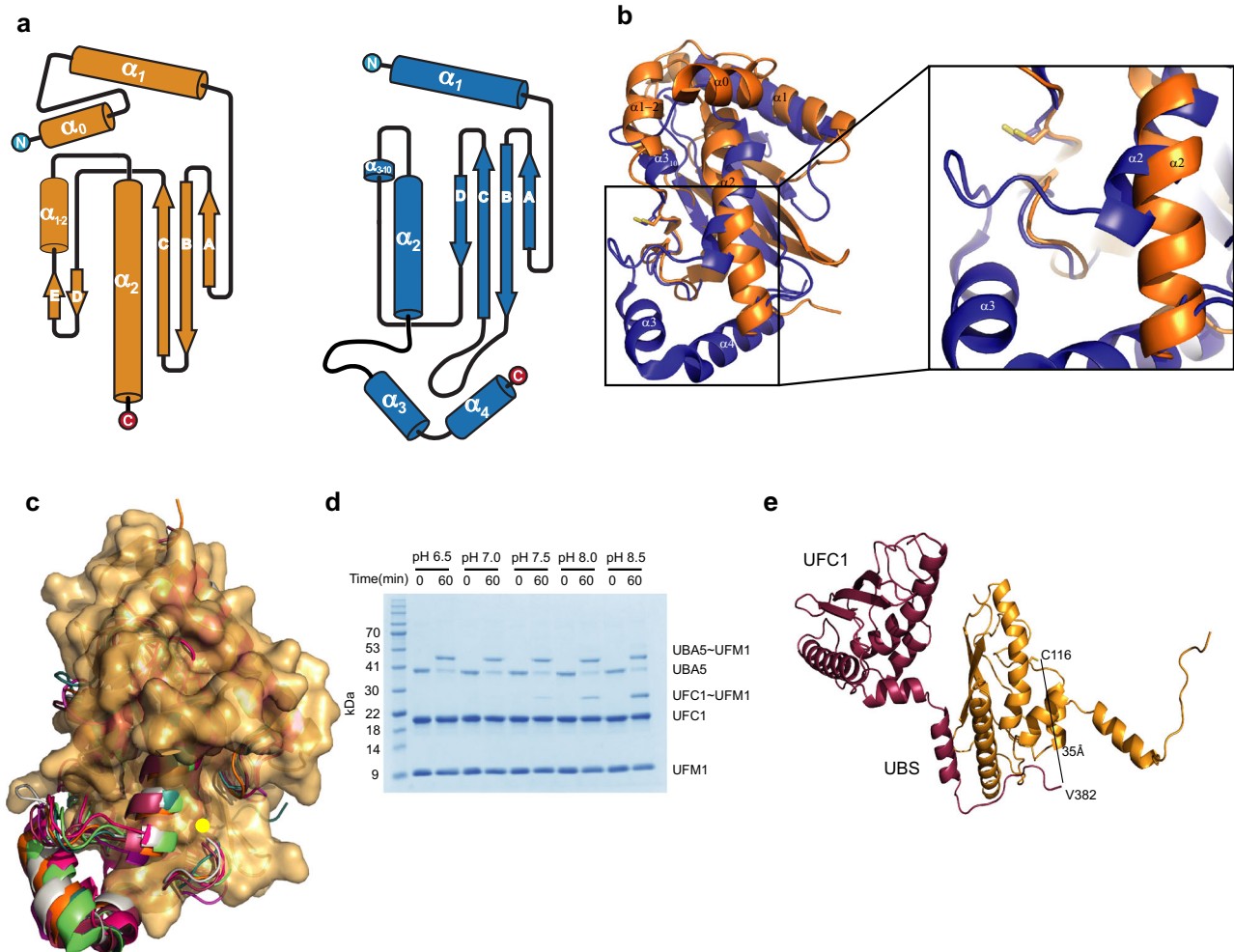

**Fig. 3 UFC1 lacks a structural loop adjacent to the active site Cys. a**. Topology depictions of UFC1 (orange) and UBC9 (blue) secondary structure elements. **b** Superposition of UFC1 (orange) with UBC9 (blue). Helices are numbered as shown in **a**. Active site Cys residues of UBC9 and UFC1 are in stick representation. **c** Superposition of UFC1 (surface representation) with seven different E2s (UBCH5b (3TGD), UBC9 (5F6E), UBC13 (1J7D), UBCH5c (1X23), E2E2 (1Y6L), UBCH7 (4Q5E), UBCH10 (4YII)) in cartoon representation. E2s are colored as shown in Supplementary Fig. 7. The yellow dot indicates the position of the active site Cys. **d** SDS−PAGE analysis showing the effect of pH on UFM1 transfer to UFC1 from UBA5 1–363. The gel is representative of at least two independent experiments. **e** Crystal structure of UFC1 fused to UBA5 residues comprising the linker and UBS (347–404). The figure shows two molecules while the UBS of the red molecule interacts with the UFC1 that arrives from the yellow molecule. Residues 347–381 of UBA5, although present in the crystal, are not shown in the structure due to a lack of electron density.

active site Cys (Fig. 4e). Next, to further support the idea that A371 reaches the active site, we investigated whether we can charge UFM1 on UFC1, employing UBA5 with the double mutation C250A and A371C. Specifically, since UBA5 residue 371 is next to the active site of UFC1, we expected that UFC1 will be able to accept UFM1 from UBA5 that is charged with UFM1 on C371. Indeed, as shown in Fig. 4f, while UBA5 C250A, as expected, failed to activate UFM1, UBA5 double mutant C250A and A371C recovered UFM1 transfer to UFC1, providing further support that it reaches the vicinity of active site cysteine. Taken together, our results suggest that the linker of UBA5 comprising residues 363-377 reaches the environment of the UFC1 active site Cys, and this probably plays a role in desolvation.

The need for an aromatic residue at position 372 of UBA5 to satisfy UFM1 transfer to UFC1 led us to hypothesize that this residue is involved in a π–π stacking interaction. To explore this possibility, we looked for UFC1 aromatic residues that are next to the active site Cys and are capable of this mode of interaction. UFC1 Y110 is located on the surface, highly exposed, and at a hydrogen-bonding distance from the thiol of C116. We first

tested whether the Y110F mutation, which keeps that aromatic ring but cannot form a hydrogen bond with C116, has any effect on UFM1 transfer. As shown in Fig. 5a, b, Y110F showed no defect in transfer, indicating that the role of this residue does not depend on hydrogen bonding with the active site Cys. Similarly, UBC9 that comprises Y87 at the same position as UFC1 Y110 is not involved in hydrogen bonding with the active site Cys (PDB 1U9A). However, when we mutated Y110 to Ala a 50% decrease in UFM1 transfer was observed (Fig. 5a, b), supporting the need for an aromatic residue at that position.

To gain structural insights into the interaction of UBA5 Y372 with UFC1 Y110, we used the Rosetta FlexPepDock peptide docking protocol[39] to model UFC1 bound to a peptide comprising UBA5 residues 370–377 (starting with blind, global docking, followed by refinement of conformations near the active site; see "Methods" section for more details). Our results converged into 2 distinct conformations of the peptide among the top-scoring models: While the conformation of the peptide C-terminal region (around K377) was homogeneous, the N-terminal part (around V370) showed a larger variance covering

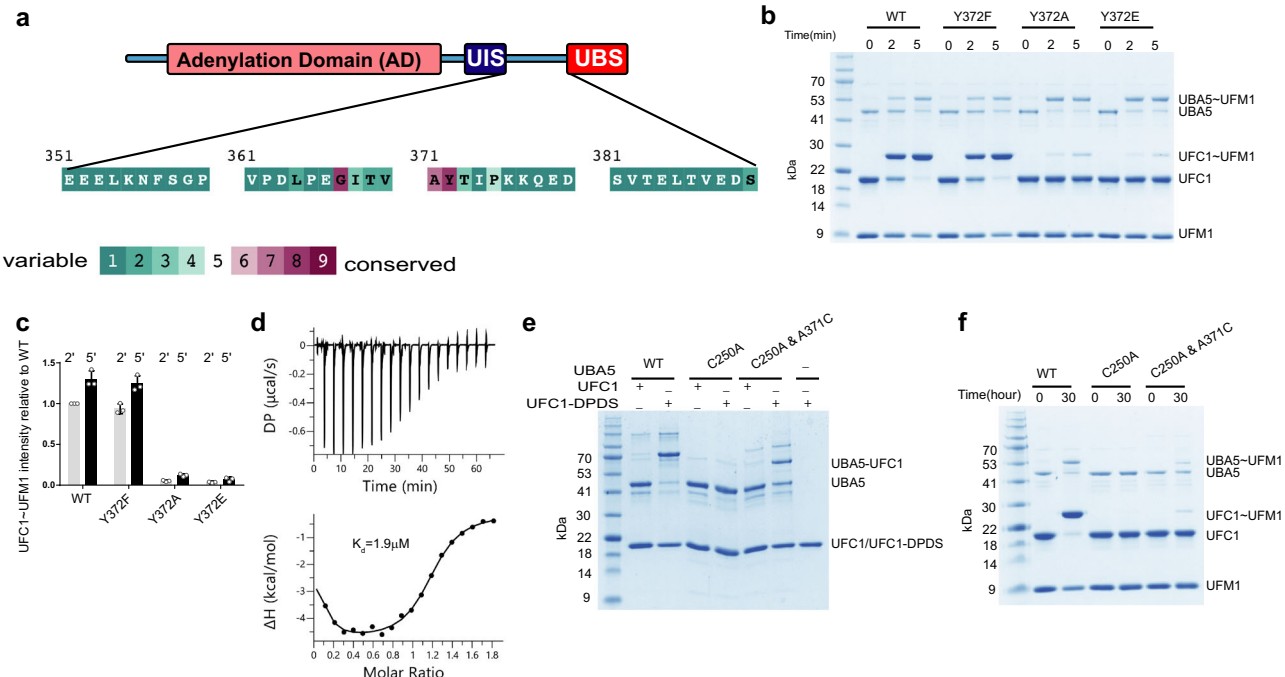

**Fig. 4 Characterization of UBA5 linker. a** ConSeq evolutionary conservation analysis of UBA5(351–390). **b** SDS–PAGE analysis showing the effect of Y372 mutations on the transfer of UFM1 to UFC1. **c** Three independent experiments as shown in **b** were performed to quantify the effect of the indicated mutations on UFM1 transfer. Data are represented by mean ± SD. **d** ITC experiment of UFC1 binding to UBA5 Y372A. The top graph represents raw data of heat flow versus time. The area under the peaks of the upper panel was integrated and plotted as kcal per mole of UFC1 as a function of binding stoichiometry in the bottom panel. Thermodynamic parameters are summarized in Supplementary Table 2. **e** UBA5 WT or mutants were incubated with UFC1 that was activated with DPDS and then were subjected to SDS PAGE analysis for UBA5-UFC1 adduct formation. The gel is representative of two independent experiments. **f** UBA5 WT or mutants were incubated with WT UFC1 and UFM1 then subjected to SDS–PAGE analysis for the formation of UFM1 charged UFC1. The gel is representative of two independent experiments.

the UFC1 interface. In one conformation, UBA5 residue Y372 is closely stacked against UFC1 Y110 (Fig. 5c), supporting π-π stacking interaction. In the second conformation (Fig. 5c), Y372 enters a cavity on UFC1's surface and points towards UFC1 F121. Indeed, F121A mutation harmed UFM1 transfer, similarly to the Y110A mutation, and a double mutant hardly showed transfer (Fig. 5a, b). However, when we increased the pH to 8.5, an improvement in the transfer was observed (Fig. 5d, e), suggesting that these mutations are not structural mutations but rather perturb the ability of UBA5 residues 363–377 to reach the active site and fulfill their role in desolvation.

To further understand how the UFC1 double mutant (Y110A and F121A) affects UFM1 transfer to UFC1, we determined its crystal structure to 2.2 Å resolution (Supplementary Table 1). As expected, the double mutant kept the overall UBC fold as observed in WT UFC1 with an RMSD of 0.3 Å². The main differences were concentrated in the loop harboring the active site Cys (Fig. 5f). Specifically, Leu 117 Cα moves 1.7 Å away from its position in the WT, and its side chain points towards the position of the missing phenylalanine (F121A). This movement is not feasible in the WT due to steric clashes with F121.

Then, to understand how these conformational changes affect UFM1 transfer, we examined our model suggesting that UBA5 Y372 utilizes a cavity on the UFC1 surface (Fig. 5c). To that end, we measured the volume of this cavity in UFC1 WT and double mutant using CASTp[40]. While in WT this cavity has a volume of 377 Å³, in the double mutant it becomes 126 Å³ (Fig. 5g). This change probably hampers the ability of UBA5 Y372 to occupy this pocket and therefore affects the recruitment of UBA5 linker to the active site Cys. Interestingly, in other E2 enzymes whose active site Cys is less exposed and therefore do not need this

mechanism of E1-mediated desolation, the corresponding cavity is significantly smaller or does not exist (Supplementary Table 4). Notably, unlike UFC1 which has the helix (α1-2), other E2s have a shorter 3₁₀ helix (Fig. 3a) that occludes the space between itself and helix 2, thereby not making room for a cavity or for only a significantly smaller one. Taken together, our results suggest that changes in the cavity volume of UFC1 due to the double mutant interferes with the ability of UBA5 to execute its role in desolvation, and accordingly, increasing the pH helps to overcome the defect in transfer.

For the desolvation, the UBA5 linker has to reach next to UFC1 active site. This prompted us to use NMR spectroscopy to define the chemical shift perturbations around UFC1 active site Cys upon binding of the linker. The reported assigned ($^1$H, $^{15}$N)-HSQC NMR spectra for UFC1[41] enables probing interactions of UFC1 at the residue level. The addition of a UBA5 fragment possessing the linker together with the UBS (residues 347–404) to $^{15}$N-labeled UFC1 caused strong attenuations at an intermediate-slow exchange on the NMR chemical shift timescale, in agreement with our determined $K_D$ of 1 μM (Fig. 6a and Supplementary Fig. 9). On the other hand, as expected, the interaction of UFC1 with UBA5 linker alone (residues 347–392) was significantly weaker, with changes occurring purely in the fast exchange regime (Fig. 6b and Supplementary Fig. 9). In line with the crystal structure of the UBA5-UBS complex, only the fragment possessing the UBS (i.e., 347–404) generated chemical perturbations in UFC1 residues that are involved in UBS binding (Fig. 6c). Both fragments perturbed residues at the N-terminal helix of UFC1 (α0) (Fig. 6a, b). Interestingly, this helix is not part of the canonical UBC fold and is missing in other E2s (Fig. 3a). This raises the possibility that α0 which was reported to provide

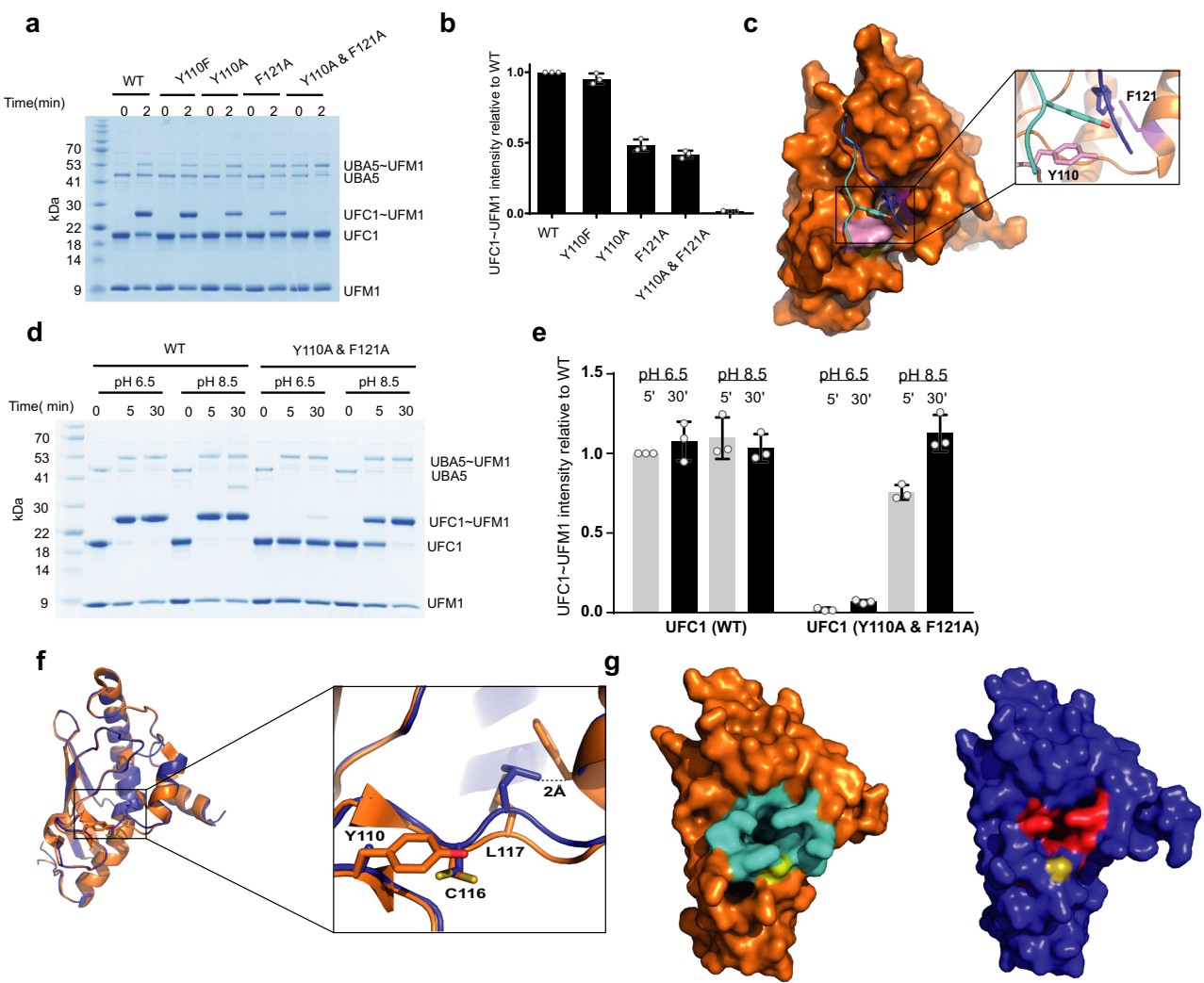

**Fig. 5 Aromatic residues are involved in UFC1 desolvation. a** SDS–PAGE analysis showing the effect of UFC1 mutations on the transfer of UFM1 to UFC1. **b** Three independent experiments as shown in A were performed to quantify the effect of the indicated mutations on UFM1 transfer. Data are represented by mean ± SD. **c** Structural model of the two main conformations (blue and cyan) of UBA5 residues 370-377 bound to UFC1. UBA5 Y372 side chain is shown in stick representation. **d** SDS–PAGE analysis showing the effect of pH on UFM1 transfer to UFC1 WT or mutant. **e** Three independent experiments as shown in **d** were performed to quantify the effect of the indicated mutations on UFM1 transfer. Data are represented by mean ± SD. **f** Superposition of UFC1 WT (orange) and UFC1 double mutant Y110A and F121A (blue). **g** Surface representation of UFC1 WT (orange) and double mutant (blue). The UFC1 residues that define the cavity are colored in cyan and red for WT and double mutant, respectively. Active site Cys is shown in yellow.

thermal stability to UFC1 may have an additional role in UFM1 transfer. Besides the N-terminal helix, the UFC1 helix (α1-2), which follows the active site Cys and exists as a short $3_{10}$ helix in other E2s, showed significant chemical perturbations with both fragments. These chemical perturbations include F121, which resides on that helix and based on our structural model approaches next to Y372 of the linker (Fig. 5c). UFC1 Y110, which is 3.5 Å from the active site Cys and possibly interacts with Y372 of the linker (Fig. 5c) is missing in the CSP analysis since it could not be assigned in our NMR experiments. However, chemical perturbation of UFC1 active site C116 was clearly observed in the presence of the fragment possessing the UBS (Fig. 6a). Overall, the NMR data support our model suggesting that the presence of the UBA5 linker alters the chemical environment of the active site Cys.

Ultimately, we aimed to assess the UBS-UFC1 structure in the context of the intact complex comprising UBA5~UFM1 and UFC1. Previously we showed that UFM1 binds UBA5 in a trans-binding mechanism[20]. In this mode of binding, UFM1 binds the UIS of one UBA5 molecule and forms a thioester bond with the

active site Cys of the other UBA5 molecule in the dimeric UBA5 (Fig. 1b). Similarly, we found that UFC1 binds the UBS of one UBA5 molecule and receives UFM1 that is linked to the active site Cys of the other molecule[20]. With these data in mind, we initially docked UFC1 (PDB 2Z6P) to UBA5-UFM1 (PDB 6H77), keeping the active site Cys of UBA5 and UFC1 next to each other. We then superimposed the structure UFC1-UBA5(389–404) onto the complex, showing that the binding of UBS to UFC1 does not block the UFC1 active site Cys from reaching the UBA5 active site (Fig. 7a). Finally, we asked whether in this ternary model A371 can reach the active site Cys as suggested by our biochemical data (Fig. 4e, f). Indeed, UFC1 active site Cys is 35 Å away from UBA5 amino acid 346, which is the last residue of the UIS. This distance can easily be filled by the 25 amino acids between residues 347–371. Accordingly, C116 is 40 Å from G391, the first residue of the UBS, a distance that can be filled by the missing 20 amino acids (390–371). Taken together, the trans binding mechanism of UFM1 and UFC1 to UBA5 is in line with our structure of UFC1-UBS and the linker that crosses the UFC1 active site Cys.

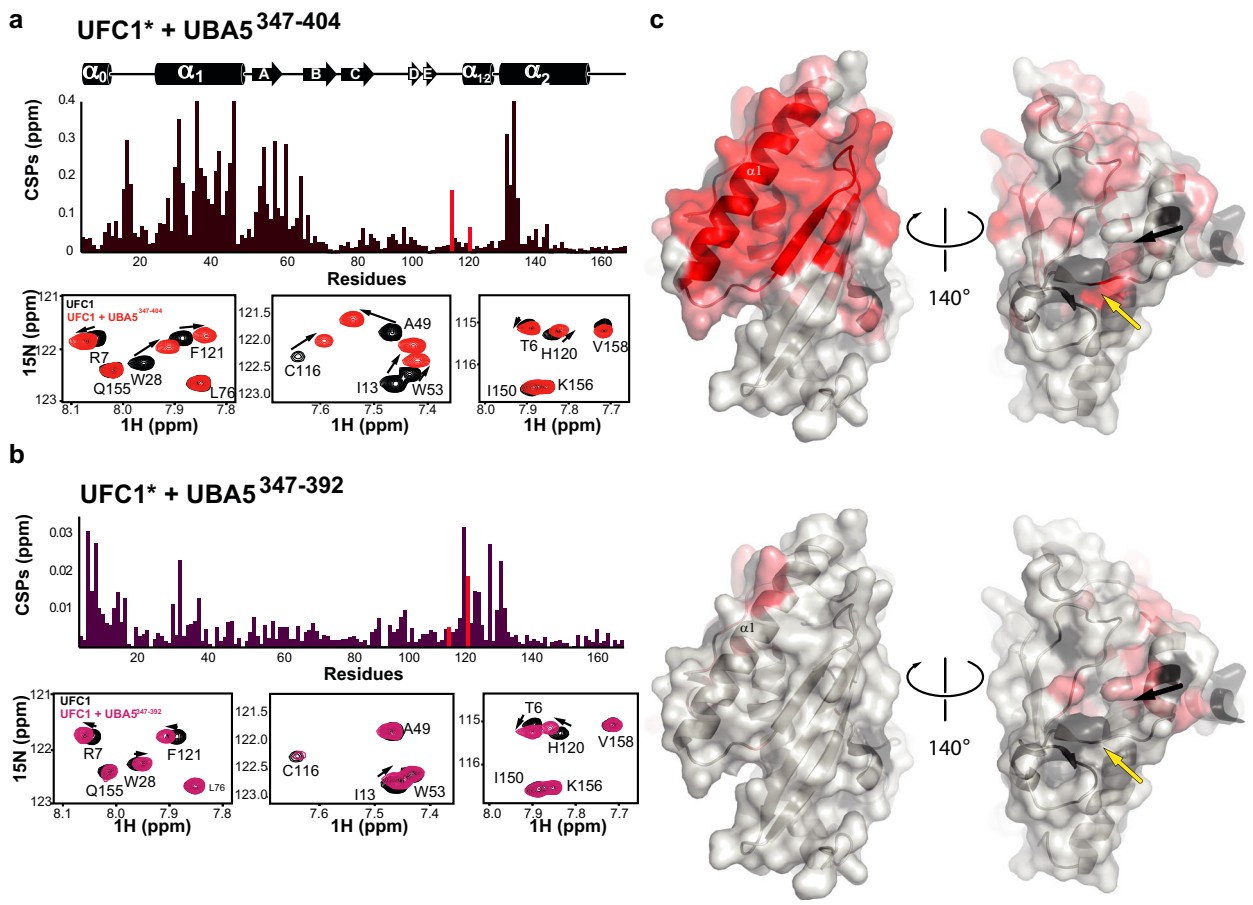

**Fig. 6 UBA5 linker affecting UFC1 active site Cys. a–b** Chemical shift perturbations (CSPs) of 0.2 mM [15]N-labeled UFC1 upon addition of equimolar concentration of UBA5[347-404] (**a**) or five molar equivalents of UBA5[347-392] (**b**). Bars correspond to UFC1 active site C116 and F121 are colored in red. Select regions of [1]H–[15]N HSQC spectra showing the spectrum of 0.2 mM UFC1 alone (black) and in the presence of an equimolar concentration of UBA5[347-404] (red; **a**, bottom row) or five-fold excess of UBA5[347-392] (purple; **b**, bottom row) **c** Structure of UFC1 (pdb 2Z6P) with the residues displaying significant CSPs (>0.04 ppm) upon interaction with UBA5[347-404] (upper panel) or UBA5[347-392] (>0.008 ppm; lower panel) colored in light red. For upper panel residues with CSPs value greater than 0.08 ppm are highlighted in red. Yellow and black arrows indicate the positions of UFC1 active site C116, or UFC1 F121, respectively.

The above ternary model proposes that UFC1 contacts the adenylation domain of both subunits of the dimeric UBA5 (Fig. 7a). One adenylation domain (shown in cartoon representation in Fig. 7a), whose active site Cys approaches the vicinity of UFC1 active site Cys, interacts with UFC1 mainly via the crossover loop. This loop harbors the active site C250 of UBA5 and as we have previously shown undergoes conformational changes upon charging of UBA5 with UFM1[20]. We expect that the interaction of UFC1 with that loop will also lead to conformational changes that are yet not clear. Specifically, the current conformation of the UBA5 crossover in our model prevents UFC1 active site Cys to reach a distance amenable for nucleophilic attack on the active site Cys of UBA5. The other adenylation domain (colored red in Fig. 7a) interacts via its N-terminus with UFC1 helix 1–2 and the loop connecting β-strands 3 and 4. As expected, the interaction of UFC1 with both adenylation domains is transient. Indeed, a binding experiment of UFC1 to UBA5 lacking the adenylation domain (UBA5 333–404) yielded a Kd that is similar to the WT protein (Fig. 7b), supporting the idea that the main region in UBA5 that is responsible for the interaction with UFC1 is outside the adenylation domain.

## Discussion

With several E1 and many E2 enzymes in the cell, the binding specificity of an E1 to its cognate E2 is crucial for the Ub/UBL conjugation process. Here we provide structural insights into the elements responsible for the binding specificity of UBA5 to UFC1. Our structures reveal that the UBS of UBA5 binds a unique pocket on the surface of UFC1, not existing in other E2s. This hydrophobic pocket is located on the other side of the active site Cys and comprises residues from α-helix 1 and β-strand 1 of UFC1. Of note, in canonical E1 enzymes a dedicated domain, the UFD, plays a role in E2 binding, and similarly to the UBS interacts with α-helix I of the UBC fold[42]. This binding of UFC1 to the UBS, in turn, brings the UBA5 active site Cys next to UFC1 active site, thereby allowing transfer of UFM1 to UFC1. Nevertheless, the UBS does not have a catalytic role in UFM1 transfer, as indicated by our ability to recover UFM1 transfer from UBA5 fragment lacking the UBS by increasing UFC1 protein concentration, supporting an affinity role.

While all E1 enzymes play the same role i.e., Ub/UBL activation and transfer to E2, UBA5 is significantly smaller in size than other E1 enzymes[42]. Accordingly, UBA5 uses a short linear sequence (UBS) for UFC1 binding while all other E1s have a domain dedicated to E2 interaction. So far, the use of a short sequence for E1–E2 interaction has been shown only from the E2 end. In detail, the E2s of the UBL NEDD8, UBE2M and UBE2F have a short sequence at their N-terminus that interacts with the heterodimeric E1, APPBP1-UBA3[43]. Currently, it is unknown whether the UBA5-UFC1 interactions are regulated in the cell.

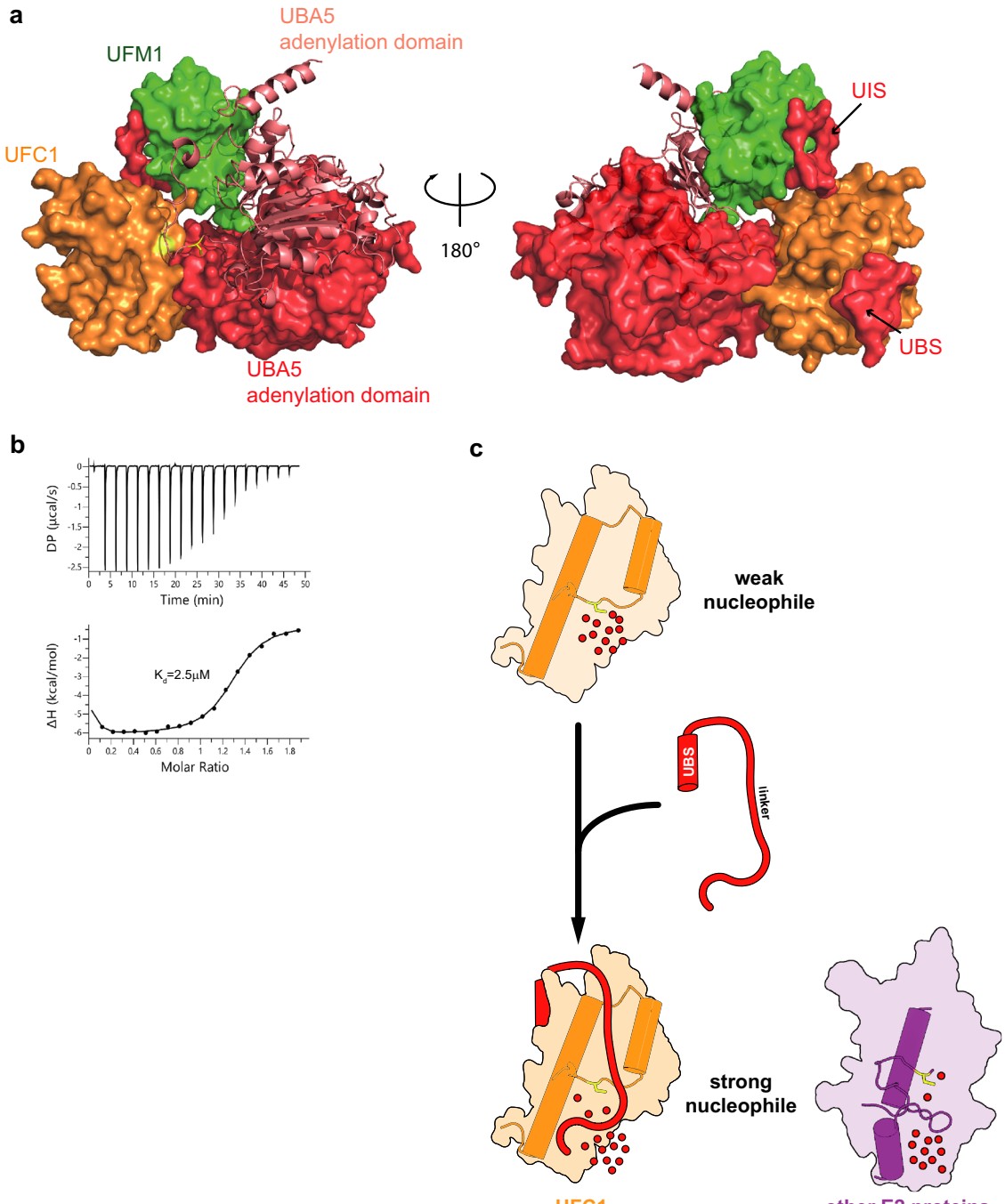

**Fig. 7 The role of UBA5 in UFM1 transfer to UFC1. a** Structural model of UBA5-UFM1 bound to UFC1. One molecule of the dimeric UBA5 is shown in surface representation (red) and the other molecule in cartoon representation (light red). UFC1 active site Cys is colored yellow and UBA5 active site C250 that approaches next to UFC1 active site is in yellow stick representation. For simplicity, the complex has only one molecule of UFC1 and the UBA5 molecule that is shown in cartoon representation has only the adenylation domain. **b** ITC experiment of UFC1 binding to UBA5 333-404. The top graph represents raw data of heat flow versus time. The area under the peaks of the upper panel was integrated and plotted as kcal per mole of UFC1 as a function of binding stoichiometry in the bottom panel. Thermodynamic parameters are summarized in Supplementary Table 2. **c** UBA5 linker plays a role in UFC1 active site Cys desolvation. The highly exposed UFC1 active site is heavily solvated by water molecules thereby reducing its nucleophilic activity. Upon binding of UBA5 via the UBS the linker approaches to UFC1 active site Cys. This reduces the amount of water molecules in the vicinity of the active site Cys and thereby elevates the latter's nucleophilic activity. In other E2s the role of the linker is executed by the loop connecting helices 2 and 3 that are missing in UFC1.

However, our structure reveals that UFC1 Y36, a known phosphorylation site[44], is located at the UBS binding interface, raising the possibility that the UFC1-UBA5 interaction can be modulated by phosphorylation.

Over the last few years, the number of reports connecting ufmylation to human diseases has significantly increased, highlighting the need for developing tools allowing specific intervention with ufmylation. Since all E1 and E2 enzymes function

with catalytic Cys, developing drugs that target the active site Cys of a specific enzyme is very challenging and can lead to a non-specific drug. Therefore, targeting additional regions on these enzymes that are unique for a specific E1 or E2 is of high interest. From that perspective, the unique mode of interaction of UBA5 and UFC1 that is mediated via the UBS can serve as a potential target for specific intervention with ufmylation. Specifically, developing peptides or small molecules that bind the UBS and prevent interaction with UFC1 could serve as specific inhibitors of ufmylation. In addition, the identified unique cavity above the active site cysteine that we propose binds Y372 of UBA5, can also serve as a potential target for developing inhibitors for ufmylation.

As noted, in contrast to other E2s, UFC1 helix 2 is longer and unbroken, causing the active site Cys to be missing a loop in its vicinity. Interestingly, sequence analysis of helix 2 cannot explain why this helix breaks in other E2s but not in UFC1. Specifically, although helix 2 in UFC1 harbors two proline residues (P144, P151), it keeps the helical structure unbroken. However, the superposition of UFC1 with UBC9 shows that a longer helix 2 in UBC9, as in UFC1, will clash with the loop connecting β-strand 2 and 3. Indeed, the superposition of E2 enzymes shows that UFC1 has a shorter loop than the other E2s. This suggests that in other E2s helix 2 breaks due to structural elements that prevent an elongated helix as seen in UFC1. How UFC1 binds the E3 ligase UFL1 is still not clear. The latter does not share any similarities with HECT or RING E3 ligases, thereby raising the question of whether the unbroken helix 2 is needed for UFC1 to function with UFL1.

UEV1a is a non-functional E2 that works together with the E2 UBE2N (Ubc13). Specifically, it binds UBE2N and holds the acceptor ubiquitin in a way that directs K63 to the thioester of UBE2N~Ub, thereby enforcing the building of K63-linked ubiquitin chains[45,46]. Unexpectedly, similar to UFC1, UEV1a ends with helix 2, thereby lacking helices 3 and 4 that exist in other E2 enzymes. This raises the question of whether UFC1 can play a similar role as UEV1a and acts together with a currently unidentified enzyme in order to build UFM1-chains. Indeed, the existence of K69-linked UFM1 chains has been reported but whether in that case UFC1 functions with another partner is not yet clear[14].

Following interaction with the E1, the E2 active site Cys has to attack the C-terminal carbonyl of Ub/UBL, breaking the thioester bond with the E1 (E1~Ub/UBL) and generating a thioester bond with the E2 (E2~Ub/UBL). This implies that the E2 active site has to be able to execute the nucleophilic attack upon binding the E1 but must not react promiscuously, which would be devastating for the cell. Previously, Tolbert et al. have suggested that the elevated pKa of E2s' active site Cys (~10) compared to free Cys (~8.5) is used to control reactivity and prevent promiscuous attacks[47]. However, here we surprisingly found that the pKa of UFC1 is significantly lower than that of other E2 enzymes, suggesting that the reactivity of UFC1 is not regulated by elevating its pKa. In contrast to ubiquitin and other UBLs that share a Di-Gly signature at the C-terminus, UFM1 ends with a Val-Gly motif. Whether Val instead of Gly challenges the nucleophilic attack thereby requiring a lower pKa for UFC1 active site Cys, which would make the latter more active, is not yet clear.

In line with its relatively low pKa, UFC1 is highly exposed to solvent compared to other E2s. While an exposed active site can benefit accessibility to substrates, it also imposes restrictions on its chemical reactivity. Specifically, the solvation of a nucleophile significantly hampers its ability to attack due to interactions with the solvent. To allow a nucleophilic attack, the highly solvated active site Cys has to undergo desolvation. In other E2s we ascribe the active site desolvation to a loop nearby the active site that is

missing in UFC1. This loop increases the buried surface area of the active site Cys, thus playing a role in desolvation. Here, we propose that in UFC1 the active site desolvation is executed by UBA5 that brings a loop next to the UFC1 active site Cys, thereby mimicking the role of the missing loop in UFC1 (Fig. 7c). This UBA5 loop, which connects the UIS to UBS, includes Y372 that occupies a cavity between Y110 and F121 of UFC1. The unperturbed UFM1 transfer process, irrespective of whether there is tyrosine or phenylalanine in the place of Y372 or Y110 of UBA5 and UFC1, respectively, suggests the essentiality of the aromatic ring rather than the hydroxyl side chain. The lesser importance of the hydroxyl group is further supported by the observation made by Yunis and Lima, regarding the UBC9 system, where they have shown that there is no perturbation in the transfer process when Y87 is mutated to phenylalanine[48]. Notably, the equivalent cavity in other E2s is highly shrunken further emphasizing that the mechanism is unique to UFC1. The coming together of three aromatic residues assists in anchoring UBA5 residues 363-377 next to the active site Cys and allows desolvation.

Taken together, ensuring binding specificity to the right E1 and controlled nucleophilic activity are two parameters critical for the proper functioning of E2 enzymes. Here we found the structural basis for how UBA5 gains its binding specificity to UFC1 via a short linear sequence located at the C-terminus. In parallel, UFC1 uses a previously unknown mechanism to regulate its nucleophilic activity. Specifically, UFC1 lacks a key structural element that is needed for its activity and is contributed by UBA5.

## Methods

**Cloning and mutagenesis.** Human UBA5 and UFM1 were cloned into pET15b and UFC1 was cloned into pET32a as previously described[20]. A UFC1 construct called UFC1-NC was generated by mutating C69 and C165 to serine leaving only the active site C116. In UBA5(Δ349-389) the missing residues were replaced by a short sequence of GSG. The fusion constructs UFC1-UBA5(389-404) and UBA5(347-404)-UFC1 were generated using Gibson assembly (Gibson assembly master mix, New England Biolabs) according to the manufacturer's protocol. The point mutants of UBA5 and UFC1 were generated by Pfu Ultra II Fusion HS DNA polymerase, an upgraded fusion polymerase technology developed by Agilent.

**Protein expression and purification.** All the proteins of UBA5, UFC1, and UFM1 that include fusion constructs, truncations, and point mutants were expressed as previously described[20]. *Escherichia coli* T7 express (New England Biolabs) expression strains were used to express all the proteins. The transformed cells were grown in 2xYT and induced at 16 °C overnight with 0.3 mM isopropyl-β-thio-galactoside (IPTG) (T-Fischer BioReagents). For NMR experiments the UFC1 transformed cells were grown in standard M9 minimal media supplemented with $^{15}NH_4Cl$ and induced at 20 °C overnight with 0.3 mM IPTG. The induced cells were harvested by centrifugation at 29,097×g for 90 min. The proteins were purified following a standardized protocol described earlier[20]. Further purification was done using 16/60 Superdex 75 pg or 16/60 Superdex 200 pg size exclusion chromatography as applicable, equilibrated in buffer containing Tris pH 7.5 (20 mM), NaCl (50 mM), and DTT (2 mM). The purified proteins were concentrated and flash-frozen in liquid $N_2$ and stored at −80 °C.

**In vitro thioester assay**

*UBA5 activation and transfer to UFC1.* UBA5 (1 μM), UFM1 (10 μM), and UFC1 (5 μM) were mixed together in a buffer containing Bis-Tris (50 mM pH6.5), NaCl (100 mM) and $MgCl_2$ (10 mM). For the reaction conducted in basic pH, Bis-Tris was replaced with HEPES (50 mM) of pH 7.0, 7.5, and 8.0. For pH 8.5, the buffer was Tris-HCl (50 mM). The concentration of the proteins was the same for all reactions involving point mutants and truncations unless specified. Reactions were initiated by the addition of ATP (5 mM) and were incubated at 30℃. A sample of the reaction was removed and quenched with SDS sample buffer supplemented with or without β-mercaptoethanol. Before activation, a sample of the reaction mix was collected as the control at time 0. The samples collected at various time points as required along with the control were then loaded on 12% Bis-Tris non-reducing SDS-PAGE and visualized by Coomassie brilliant blue R staining.

*Trans addition of UBA5 C-terminus to UBA5 1-347.* The C-terminus of UBA5 347 to 404 (60 μM) was added to rescue the loss of function of UBA5 1–347. The concentrations of the proteins were as described in the above paragraph. Reactions were initiated by the addition of ATP (5 mM) and were incubated at 30 °C. The samples were collected twice, after 30 min and after 1 h. The reaction was quenched

with SDS sample buffer without BME. Before the addition of ATP, a sample of the reaction mix was collected as a control. The samples were then loaded on 12% Bis-Tris non-reducing SDS-PAGE and visualized by Coomassie brilliant blue R staining.

**Free cysteine discharge assay.** UBA5 1–347 (1 μM) and UFM1(10 μM) were mixed together in a buffer containing Bis-Tris (50 mM pH 6.5), NaCl (100 mM), and MgCl₂ (10 mM). Another reaction employing UBA5-WT was carried out in parallel as a control. A sample of the reaction mixture was collected before the start of the reaction. Reactions were initiated by the addition of ATP (5 mM) and were incubated at 30 °C. Two samples of the reaction mixture were collected after 10 min incubation and quenched with SDS sample buffer with and without β-Mercaptoethanol. Subsequently, free cysteine (100 mM) was added to the reaction mixture and incubated at 30 °C. A sample of the reaction mix was collected after 10 min and quenched with SDS sample buffer without β-Mercaptoethanol. The samples were then loaded on 12% Bis-Tris non-reducing SDS-PAGE and visualized by Coomassie brilliant blue R staining.

**Isothermal titration calorimetry (ITC).** Experiments were carried out on a MicroCal PEAQ ITC system (Malvern Instruments, Malvern) at 25 °C. The binding experiments of UBA5(Δ349–389), and point mutants UBA5 M401R, L397R, and Y372A to UFC1-WT were conducted in buffer containing Tris (20 mM pH 7.5), NaCl (50 mM), and DTT (2 mM). Measurements were obtained from 19 injections where each injection is 2 μl. The data were fitted to the two sets of sites model already built into the MicoCal PEAQ ITC analysis software. The initial injection volume in all experiments was 0.4 μl with a duration of 0.8 s. Data for the first injection were not considered in any experiments.

**Crosslinking of UFC1-NC with UBA5 constructs.** UFC1-NC was activated with the crosslinker compound 2,2' dithiodipyridyldisulfide (DPDS) as described[23]. The protein was desalted to remove DTT present in the storage buffer. About 100–200 μM of UFC1-NC were treated with 2.5 mM of the crosslinker and incubated at 25 °C for 30 min. Then, the activated UFC1-NC was desalted to remove the excess crosslinker. Activated UFC1-NC (4 μM) was mixed with 2 μM of UBA5 (WT or mutants) in PBS buffer at 25 °C for 20 min. A sample of the reaction mix was removed and quenched with SDS sample buffer without β-mercaptoethanol and loaded on 12% Bis-Tris non-reducing SDS-PAGE and visualized by Coomassie brilliant blue R staining.

**Crystallization.** All crystals were grown at 20 °C using the hanging drop vapor diffusion method. UFC1-UBA5(389–404) protein was concentrated to 15 mg/ml and crystallized in the solution containing 35 mM citric acid, 65 mM bis-tris propane, 19% PEG3350, 100 mM lithium chloride. The quality of the crystals was improved by including lithium chloride as an additive. The crystals were soaked in a 30% glycerol cryo mixture and flash-frozen in liquid nitrogen. UBA5(347–404)-UFC1 at concentration of 34 mg/ml was crystallized in the solution containing 2% (v/v) tacsimate pH 7.0, 20% PEG 3350, 0.1 M HEPES pH 7.5, 6 mM zinc sulfate. The crystals appeared after 2 days and grew to a considerable size in 4 days. The crystals were cryo-protected using a paraffin-paratone mixture and flash-frozen in liquid nitrogen. UFC1 Y110A and F121A double mutant at a concentration of 12 mg/ml was crystallized in a solution containing 0.1 M Hepes pH 7.5 and 2 M ammonium sulfate. The crystals appeared within 24 h. The crystals were cryo-protected using 18% glycerol.

**X-ray data collection and processing.** The diffraction data for the UFC1-UBA5(389–404) were collected at the European Synchrotron Radiation Facility on beamline ESRF ID23 at 100 K in a stream of gaseous nitrogen. Data were processed using XDS[49]. The orthorhombic data demonstrated strong anisotropy and were therefore subjected to anisotropic ellipsoidal truncation using the STARANISO server[50] (Supplementary Table 1). UBA5(347-404)-UFC1 data were collected on beamline BL14.1 of BESSY-II synchrotron at 100 K. Data were processed using XDS (Supplementary Table 1). UFC1 Y110A and F121A diffraction data were collected on XtalLab Pro (Rigaku) with a PILATUS 200K detector. Data were processed using HKL 3000 suite (Supplementary Table 1).

**Structure determination.** The structure of the UFC1-UBA5(389–404) protein was solved by molecular replacement method (MR) with MOLREP[51] using a high-resolution structure of UFC1 (PDB 2Z6O)[32]. The asymmetric unit contained two monomers of the fusion protein. The partial model was initially refined in REFMAC5[52] and electron density maps were inspected in COOT[53]. Helical difference density was observed close to the same regions of both monomers of UFC1. Phased molecular replacement in MOLREP[54] positioned a ten-residue ideal α-helical polyalanine fragment into the observed density. To assign side chains of the UBA5 fragment, the electron density was averaged using DM[55] and the model was refined in REFMAC5 with input density modification phases[56]. In addition, the model was subjected to refinement in BUSTER[57]. The resulting maps allowed confident assignment of the protein sequence for the UBA5 fragment.

The structure of UBA5(347–404)-UFC1 was solved by the MR method implemented in MOLREP using the solved structure of UFC1-UBA5(389–404).

The model was refined in REFMAC5 and subsequently remodeled in COOT. Multi crystal averaging using the UBA5(347–404)-UFC1 map with the map of the high-resolution native UFC1 structure was performed using DMMULTI[55]. The structure was refined in both BUSTER and REFMAC5 with averaged phases. The electron density maps confirmed amino acid assignment for stretch 389–404 and additionally allowed to build backward the model of the UBA5 fragment up to Val382 in COOT. The residues from 347 to 381 of the UBA5 fragment were not built due to the lack of interpretable electron density. The structure of UFC1 Y110A and F121A was solved by the MR method implemented in the PhaserMR module of CCP4 suite[58] using the native UFC1 structure (PDB 2Z6P). The model was refined in REFMAC5 and subsequently remodeled in COOT.

**NMR spectroscopy.** All NMR experiments were carried out at 25 °C on a 23.5 T (1000 MHz) Bruker spectrometer equipped with a triple resonance (x,y,z) gradient cryoprobe. The experiments were processed with NMRPipe[59] and analyzed with NMRFAM-SPARKY[60].

The interaction of UFC1 with UBA5 fragments was monitored by 2D ¹H–¹⁵N HSQC experiments with the assignments for UFC1 transferred from the BMRB (entry 6546). UBA5³⁴⁷⁻⁴⁰⁴ (50–200 μM) or UBA5³⁴⁷⁻³⁹² (0.4–1.6 mM) were titrated into 200 μM of ¹⁵N-labeled UFC1 in 50 mM HEPES pH 7.0, 50 mM KCl, 1 mM DTT, 0.03% NaN₃, and 10% D₂O and chemical shifts were recorded.

CSPs were calculated from Eq. (1):

$$\Delta\delta = \sqrt{\Delta\delta_H{}^2 + \left(\frac{\Delta\delta_N}{5}\right)^2} \tag{1}$$

where $\Delta\delta_H$ is the amide proton chemical shift difference and $\Delta\delta_N$ is the ¹⁵N backbone chemical shift difference. CSPs measured with UBA5³⁴⁷⁻⁴⁰⁴ and UBA5³⁴⁷⁻³⁹² fragments were considered significant if greater than half standard deviation from the mean (>0.04 ppm or >0.008 ppm, respectively), with residues displaying CSPs greater than 0.2 excluded from the mean calculations.

**Theoretical estimation of pKa of the catalytic cysteine.** The theoretical pKa and the buried surface area of the active site cysteine residue in all the E2 proteins including UFC1 were estimated using the PDB2PQR module in Applied Poisson Boltzmann Server (APBS)[61]. The server estimates the continuum electrostatics for large biomolecular assemblage using the Poisson-Boltzmann equation. The coordinates of the protein structure after removing the solvents and water molecules were uploaded to the server.

**Ternary complex model.** The structure of UFC1-UBA5(389–404) was docked to UBA5-UFM1 structure (PDB 6H77) using Patchdock[62]. The UBA5-UFM1 structure was used as the receptor while the UFC1-UBA5(389–404) fused structure was used as the ligand. A distance constraint of 5 Å between C116 of UFC1 and C250 of UBA5 was applied for docking. The receptor-binding site parameter file constitutes 5 residues upstream and 5 residues downstream of C250 of UBA5. Likewise, the ligand-binding site parameter file constitutes UFC1 residues from Y110 to H120, including C116.

**Docking of UBA5 linker to UFC1 using Rosetta FlexPepDock peptide docking.** Global docking with no prior information about the peptide conformation and the exact binding site was performed using the PIPER-FlexPepDock protocol (PFPD)[63], and then further refined with FlexPepDock refinement (FPD)[39]. UBA5 segment 370-VAYTIPKK-377 was used for docking instead of the whole linker sequence. A shorter segment was picked under the assumption that this region encompasses the crucial part of the interaction, and since fragment representation as used in PFPD is more restricted for longer peptides (hence less optimal). The UFC1-UBA5(389–404) structure was used as the receptor. The top 500 models (out of 12,500, by FPD reweighted score) were examined. Conformations that located the peptide within ~7 Å from UFC1 active Cys 116 and were compatible with connection to V382 were selected for further refinement, including both low-resolution and high-resolution steps in FPD (1000 samples).

**Cavity volume estimation.** Cavity volume calculation and topography mapping of the E2 PDB structures were performed using the computed atlas of surface topography of proteins (CASTp)[40]. All solvent molecules including water molecules were removed before the calculation was made. CastP lists all the cavities present in the protein with details of volume, area, and residues present in the cavity. The probe radius was set at the default value of 1.4 Å for calculation. The cavities were visualized and analyzed using PyMOL.

**Reporting summary.** Further information on research design is available in the Nature Research Reporting Summary linked to this article.

## Data availability

Atomic coordinates and structure factors were deposited in the RCSB PDB (https://www.rcsb.org/) with the accession codes 7NW1, 7NVK, and 7NVJ for UFC1-UBA5 (389–404), UBA5(347-404)-UFC1, and UFC1(Y110A and F121A), respectively. NMR

assignments for UFC1 were taken from the BMRB entry 6546. Previously published crystal structures used in this study are available from the RCSB PDB under the accession codes: 3TGD; 1J7D; 1U9A; 1×23; 1Y6L; 4Q5E; 4YII; 1Y8X; 1WZW; 6CYO; 1FZY; 1YLA; 2YBF; 2C4P; 5LBN; 3FN1; 2CYX; 2Z5D; 2F4W; 5BNB; 1YH2; 1YRV; 2Z6P; 2Z6O; 1JBB; 4Q5H; 1WZV; 3RZ3; 2DYT; 6H77. The coordinates of the structural models generated by in silico docking are provided as Supplementary Data 1–3. Source data are provided with this paper.

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

## Acknowledgements

We thank the beamline staff of BESSY-II 14.1 and ESRF ID23. This work was supported by the Israel Science Foundation (grant 1383/17) to R.W, (grant 1889/18) to R.R, (grant 717/2017) to O.S.F and (grant 401/18) to M.D, the Israel Cancer Research Fund (grant 3013000281) to R.W and the US-Israel Binational Science Foundation (grant 2015207) to O.S.F.

## Author contributions

M.K. and R.W. designed the experiments; M.K. P.P., and J.F. performed the biochemical experiments; S.B. and E.C.K. performed the cloning; M.K. P.P., and F.H. carried out the protein purification; T.T. and O.S.F. modeled the structure of UFC1 with UBA5 linker; G.Z. and R.R. designed and performed the NMR experiments; M.K. and P.P. grew the crystals and collected the crystallographic data together with M.D. and R.W; M.K., P.P., M.I., and R.W. determined the crystal structures; M.K., T.T., M.I., R.R., O.S.F., and R.W. wrote the manuscript.

## Competing interests

The authors declare no competing interests.

## Additional information

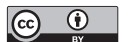

