## [Peer Review File · Nature Communications]

REVIEWER COMMENTS

Reviewer #1 (Remarks to the Author):

Key results (summary)

In their manuscript entitled “Structural basis for UFM1 transfer from UBA5 to UFC1” Kumar et al., present the mechanism of the UBA5 mediated UFM1 transfer to the E2 enzyme UFC1. These findings are important to understanding UFM1 enzyme activity as they shed light into the previously unknown structural mechanism that directs UFM1 transfer directly and specifically to UFC1 through a novel interaction between UBA5 and UFC1. These findings significantly impact the structural and biochemical understanding of the E2 enzyme UFC1 by giving insights into the UFM1 transfer mechanism.

Originality and significance:

This work represents an advancement in the structural work for the E2 enzyme UFC1, the only articles describing its structure were published more than a decade ago[1, 2]. Furthermore, this manuscript reports of a unique interaction between UBA5 and UFC1 regulating UFM1 transfer and describes a novel mechanism for ensuring UFC1 specificity. Taken together, these results are significant for understanding UFMylation, as they shed light into underlying the structural and biochemical mechanisms.

With some modifications, in particular textual improvements, as well as some additional analysis that are outlined below, this manuscript could be suited for publication in Nature Communications. In particular, the novelty of these structural mechanisms and the significance of these findings for UFMylation research warrant its publication in this journal. However, to really elevate the message of their discoveries, the authors should take the experimental suggestions into consideration.

Suggested improvements

1. Textual aspects

The abstract could stress the novelty of the authors findings more given this is an important novel discovery shedding light into how UBA5 and UFC1 cooperatively activate and transfer UFM1. Underscoring that this discovery is key to understanding the unique catalytic activity of UFC1 and deciphering the UFMylation system but also opens up new avenues for inhibitor development.

Similarly, the conclusion should highlight the impact of the author's findings. For example, would it be conceivable that this novel mechanism that UFC1 uses to regulate its nucleophilicity might allow the transfer of UFM1 directly to substrates without the need of an E3 ligase UFL1, as is the case for several Ubiquitin E2 enzymes[3]? The authors describe that UFC1 possess a novel mechanism to regulate its nucleophilic activity rendering it highly specific for UFM1. I was wondering whether the pKa of the UFC1 active site cysteine is necessary to allow a nucleophilic attack on the UFM1 thioester intermediate, since UFM1 contains a C-terminal VG-motif, instead of the di-Glycine motif found in Ubiquitin and other Ubiquitin-like modifiers. Also, would the lower pKa value of the UFC1 active site cysteine prevent the enzyme from reacting with LC3, which also has a VG-motif at its C-terminus? I believe this would be an interesting discussion and would further underscore the specificity of UFC1 for UFM1.

In the discussion, the authors mention that “the non-functional E2 variant UEV1A, which has an E2 fold, also lacks the loop”. This is an interesting observation and raises the questions, whether UFC1 might have a similar role as UEV1A, which functions as a co-factor for UBC13—an E2 enzyme building K63 Ubiquitin chains[4]. It would strengthen the discussion if the authors would mention that UEV1A is a co-factor of UBC13 and perhaps expand if UFC1 might function—together with a currently unidentified enzyme—in poly-UFMylation, as suggested by Yoo et al.[5]. Furthermore, the results presented in this manuscript might permit the hypothesis that the unique structural features of UFC1 are required for interaction with the ligase UFL1, which although currently not structurally characterized—does not share any similarities with the Ubiquitin E3 ligases.

Lastly, I believe the discoveries presented here have the potential impact the discovery and development of inhibitors (for UBA5 or UFC1), by providing alternative sites other than the active site cysteine for pharmacological intervention. For example, it would be conceivable that an inhibitor targeting the UBS domain of UBA5 would disrupt the UBA5-UFC1 interaction and therefore UFM1 transfer to the substrate. The authors should perhaps take this into consideration in their discussion.

2. Figures

Figure 1: Including a schematic illustrating the structural relationship of UFC1 and UBA5, similar to the schematic depicted in the graphical abstract in Oweis et al.[6], would be helpful to the reader to visually aid the understanding of the mechanism.

Figure 2: Space between the line separating the time and the variant labels needs to be adjusted as it is not consistent with the one in subfigures C and D. Also, a quantification of the “kinetics” of UFM1 transfer would be useful. In some cases, strong impairment of thioester formation is visible on the gel, but in other cases the difference is not readily seen. Including some type of quantification of the gel-based assays would be helpful. This also applies for the gels in Figures 4 and 5.

Additionally, a schematic diagram of UBA5 binding to UFC1 would help in understanding the model, similar to the one in Oweis et al.,[6]. Also including the linker region in this schematic would graphically depict the observations making it easier to understand.

Figure 4 C): This figure is not readily understandable without reading the figure legend first. It could be improved by labeling it with UBA5 and perhaps including a scheme of the location UBA5 in relation to this region (e.g. zoom in on the structure to show the location of this region in regard to the entire protein).

Figure 5: In the cartoon representation of the superimposition of UFC1 with seven different E2 enzymes, it would be advisable to include a legend in the figure with the color of each E2 enzyme. Also, providing a sequence alignment of the different E2 enzymes they used for the model (especially the colored helix region) could be included (perhaps as a supplementary figure).

Figure legends: Throughout the figure legends the term “SDS-PAGE” has been used, while it is obvious that the authors most likely meant “SDS-PAGE gels” or “SDS-PAGE gel analysis”. This should be corrected. Also, the naming of the double mutant Y110 & F121A should be written as Y110A and F121 when referring to it in the text.

3. Experimental Suggestions:

To obtain the crystal structure, the authors generated a UBA5-UFC1 fusion construct. Could the authors elaborate the rationale behind this decision? Both UBA5 and UFC1 can be easily expressed in *E. coli* and could be added to UFM1 in the presence of ATP and MgCl₂ to generate a UFC1 thioester intermediate. Given that the thioester intermediates (Uba5-UFM1 and UFC1-UFM1) are unstable and difficult to crystalize, would it be feasible to use chemical probes [7, 8] to generate a stable thioester mimic in complex with UBA5? Furthermore, can the helix 2 of UFC1 be modified by the use of linkers (for example linkers similar to the ones reported in [9]) to support the modelling data? This approach, if structural data can be obtained, would provide further structural insights on the UFM1 thioester transfer from the E1 (UBA5) to the E2 enzyme (UFC1). Furthermore, comparison to existing literature [10] could be made.

Just as a thought-provoking question: Would it be possible to generate an E2 enzyme in which the loop region (shown in Figure 3B) is deleted and then test if this E2 enzyme can function with UBA5 to charge UFM1?

4. References:

1. Liu, G., et al., NMR and X-RAY structures of human E2-like ubiquitin-fold modifier conjugating enzyme 1 (UFC1) reveal structural and functional conservation in the metazoan UFM1-UBA5-UFC1 ubiquitination pathway. *J Struct Funct Genomics*, 2009. 10(2): p. 127-36.
2. Mizushima, T., et al., Crystal structure of Ufc1, the Ufm1-conjugating enzyme. *Biochem Biophys Res Commun*, 2007. 362(4): p. 1079-84.
3. Stewart, M.D., et al., E2 enzymes: more than just middle men. *Cell Res*, 2016. 26(4): p. 423-40.
4. Petroski, M.D., et al., Substrate modification with lysine 63-linked ubiquitin chains through the UBC13-UEV1A ubiquitin-conjugating enzyme. *J Biol Chem*, 2007. 282(41): p. 29936-45.
5. Yoo, H.M., et al., Modification of ASC1 by UFM1 is crucial for ERalpha transactivation and breast cancer development. *Mol Cell*, 2014. 56(2): p. 261-274.
6. Oweis, W., et al., Trans-Binding Mechanism of Ubiquitin-like Protein Activation Revealed by a UBA5-UFM1 Complex. *Cell Rep*, 2016. 16(12): p. 3113-3120.
7. Mulder, M.P., et al., A cascading activity-based probe sequentially targets E1-E2-E3 ubiquitin enzymes. *Nat Chem Biol*, 2016. 12(7): p. 523-30.

8. Witting, K.F., et al., Generation of the UFM1 Toolkit for Profiling UFM1-Specific Proteases and Ligases. *Angew Chem Int Ed Engl*, 2018. 57(43): p. 14164-14168.
9. Kamadurai, H.B., et al., Mechanism of ubiquitin ligation and lysine prioritization by a HECT E3. *Elife*, 2013. 2: p. e00828.
10. Page, R.C., et al., Structural insights into the conformation and oligomerization of E2~ubiquitin conjugates. *Biochemistry*, 2012. 51(20): p. 4175-87.

Reviewer #2 (Remarks to the Author):

In this manuscript, Kumar and Padala et al. focused their efforts on understanding basic mechanisms of the E1/E2/E3 enzymatic machinery responsible for protein UFMylation. The authors specifically zeroed in on an early step of the UFM conjugation cascade involving: 1) recruitment of the E2 (UFC1) to the E1 (Uba5) and 2) transfer of activated UFM1 from E1 to E2. To address these issues, the authors determined crystal structures of E2 in complex with two different E1 fragments and complemented their structural findings with a number of biochemical, biophysical, and molecular modeling studies. Together, these structures reveal the molecular basis for UBS-mediated recruitment of UFC1 to UBA5 and uncovered a surprising and unexpected role for a stretch of residues located between the UIS and UBS in promoting catalysis of transthioesterification that the authors propose functions through a desolvation mechanism. Finally, the authors used the insights gained from their collective studies to construct a model of the Uba5-UFC1-UFM1 complex in a catalytically active state. Overall, the manuscript is concise and the biochemical experiments in particular are elegant and carefully executed. The study is significant because it potentially fills in several key steps in our understanding of E1-E2 transthioesterification in the UFM1 system. With that said, there are some issues that should be considered and addressed to warrant publication in *Nature Communications*:

Key points:

The Uba5 fragment 363-377 is proposed to play a key role in transthioesterification by desolvating the E2 catalytic cysteine. However, this region is disordered in the crystal structure harboring the 347-404 fragment of Uba5. It is reasonable to assume that this could be the result of a transient interaction and the authors did perform some nice biochemistry to test their proposed role for the 363-377 fragment in desolvation of the E2 catalytic cysteine. However, given the key role of this

interaction to the significance and novelty of the study I wonder if the authors could perform additional experiments that would provide deeper molecular/structural insights into the mechanism. For example, the NMR resonance assignments of UFC1 are available (BMRB Entry 6546) and it seems like it would be relatively straightforward to perform CSP experiments to map the UFC1 residues involved in the interaction with the 363-377 Uba5 fragment. A reciprocal experiment could map the Uba5 residues involved in this interaction. A second approach to gaining more insight to this process would be to leverage the A371C mutant the authors demonstrated was capable of cross-linking to UFC1 catalytic cysteine by attempting to crystallize a A371C mutant of the fusion protein they determined the structure of in this study. Presumably the disordered residues were present in the crystal which suggests that the crosslinked A371C species may crystallize in the same space group in a similar crystallization condition. Have the authors considered such studies?

In the model of the Uba5-UFC1-UFM1 complex presented in Figure 5G are there any new interactions between the E1 and E2 active sites that are observed or any predicted interactions between the 363-377 Uba5 fragment and UFC1 that play a role in complex formation and catalysis? If more insight could be gained from this model that could be tested in functional studies it could increase the significance of the study significantly.

Minor points:

*Please show composite omit map density for the UBS in both complex structures in the supplementary material

*Do the authors have a sense of why the UBS interacts with UFC1 in 'trans' in the AU of both fusion constructs or is this just coincidence?

*Many of the figures could be more clearly labeled to help readers to interpret them. Please label Fig. 1B to highlight the active site more clearly. Please also label secondary structure elements of E2

*Fig 2D- why is the molecular weight of the Uba5 347-404 fragment being added in trans so much higher than what would be expected? If there is a tag that remains?

*page 13, table 3- these are predicted pKa values, correct? If so, this should be stated.

*please more thoroughly label figure 4B

*page 16 line 6- Fig 4A&B should be figure 5.

*Figure 5G is confusing to me... based on Oweiss et al (Cell Reports 2016) UFM1 an UFC1 binding and transthioesterification occur in trans. In this figure it appears that it is only one protomer of the Uba5 dimer that is being shown? Overall this version of the figure should be more clearly labeled. Should the UBS be red? Also the authors should consider a cartoon model for the mechanism of transthioesterification and how the new findings in this paper fit into our understanding of this process.

REVIEWER COMMENTS

Reviewer #1 (Remarks to the Author):

Key results (summary)

In their manuscript entitled “Structural basis for UFM1 transfer from UBA5 to UFC1” Kumar et al., present the mechanism of the UBA5 mediated UFM1 transfer to the E2 enzyme UFC1. These findings are important to understanding UFM1 enzyme activity as they shed light into the previously unknown structural mechanism that directs UFM1 transfer directly and specifically to UFC1 through a novel interaction between UBA5 and UFC1. These findings significantly impact the structural and biochemical understanding of the E2 enzyme UFC1 by giving insights into the UFM1 transfer mechanism.

Originality and significance:

This work represents an advancement in the structural work for the E2 enzyme UFC1, the only articles describing its structure were published more than a decade ago[1, 2]. Furthermore, this manuscript reports of a unique interaction between UBA5 and UFC1 regulating UFM1 transfer and describes a novel mechanism for ensuring UFC1 specificity. Taken together, these results are significant for understanding UFMylation, as they shed light into underlying the structural and biochemical mechanisms.

With some modifications, in particular textual improvements, as well as some additional analysis that are outlined below, this manuscript could be suited for publication in Nature Communications. In particular, the novelty of these structural mechanisms and the significance of these findings for UFMylation research warrant its publication in this journal. However, to really elevate the message of their discoveries, the authors should take the experimental suggestions into consideration.

We would like to thank the reviewer for his/her comments that helped us to significantly improve our manuscript.

Suggested improvements

1. Textual aspects

The abstract could stress the novelty of the authors findings more given this is an important novel discovery shedding light into how UBA5 and UFC1 cooperatively activate and transfer UFM1. Underscoring that this discovery is key to understanding the unique catalytic activity of UFC1 and deciphering the UFMylation system but also opens up new avenues for inhibitor development.

We have edited the abstract according to the reviewer’s suggestions.

Similarly, the conclusion should highlight the impact of the author’s findings. For example, would it be conceivable that this novel mechanism that UFC1 uses to regulate its nucleophilicity might allow the transfer of UFM1 directly to substrates without the need of an E3 ligase UFL1, as is the case for several Ubiquitin E2 enzymes[3]?

To follow this idea that UFC1 can function without UFL1, we tested whether we can discharge UFC1~UFM1 with free Lys. As shown in the figure below, we did not get

transfer to free Lys but only to free Cys. In this experiment the high concentration of free Lys mimics Lys of substrates. While this experiment does not rule out the possibility that UFC1 cannot transfer directly to Lys on substrate, it supports an intermediate step in which UFM1 is transferred to Cys on E3 prior to the formation of an isopeptide bond. Furthermore, in UBC9 and other E2s, the loop that is missing in UFC1 plays a role in activating the attacking Lys (Yunus and Lima, NSMB 2006), providing further support to the idea that UFC1 does not transfer UFM1 directly to the substrate.

UFC1 discharge by different nucleophiles. Free Cys or hydroxylamine (NH₂OH), but not free Lys, can discharge UFM1 from UFC1. All nucleophiles are at 50 mM concentration.

The authors describe that UFC1 possess a novel mechanism to regulate its nucleophilic activity rendering it highly specific for UFM1. I was wondering whether the pKa of the UFC1 active site cysteine is necessary to allow a nucleophilic attack on the UFM1 thioester intermediate, since UFM1 contains a C-terminal VG-motif, instead of the di-Glycine motif found in Ubiquitin and other Ubiquitin-like modifiers.

We have now raised this point in the discussion (page 16).

Also, would the lower pKa value of the UFC1 active site cysteine prevent the enzyme from reacting with LC3, which also has a VG-motif at its C-terminus? I believe this would be an interesting discussion and would further underscore the specificity of UFC1 for UFM1.

This is an interesting point; however, as we previously showed, UFM1 R81 is needed for activation by UBA5 (Oweis et al. Cell Reports 2016). The corresponding residue in LC3 is Thr, suggesting that LC3 will not work with the ufmylation machinery. Also, LC3 lacks the canonical GG motif existing in Ub and many UBLs. Instead, LC3 ends with FG. This aromatic residue instead of the Val in UFM1 may also prevent from LC3 to work with the UFM1 machinery.

In the discussion, the authors mention that “the non-functional E2 variant UEV1A, which has an E2 fold, also lacks the loop”. This is an interesting observation and raises the questions, whether UFC1 might have a similar role as UEV1A, which functions as a co-factor for UBC13—an E2 enzyme building K63 Ubiquitin chains[4]. It would strengthen the discussion if the authors would mention that UEV1A is a co-factor of UBC13 and perhaps expand if UFC1 might function—together with a currently unidentified enzyme—in poly-UFMylation, as suggested by Yoo et al.[5]. Furthermore, the results presented in this manuscript might permit the hypothesis that the unique structural features of UFC1 are required for interaction with the ligase UFL1, which although currently not structurally characterized—does not share any similarities with the Ubiquitin E3 ligases.

Following the reviewer suggestions, we have elaborated on UEV1A. In the discussion we raise the possibility that the unique structural elements are needed for UFC1 to function with UFL1. In addition, we discuss the possibility that UFC1 also functions as an auxiliary subunit for another unknown E2 (page 15 in the discussion).

Lastly, I believe the discoveries presented here have the potential impact the discovery and development of inhibitors (for UBA5 or UFC1), by providing alternative sites other than the active site cysteine for pharmacological intervention. For example, it would be conceivable that an inhibitor targeting the UBS domain of UBA5 would disrupt the UBA5-UFC1 interaction and therefore UFM1 transfer to the substrate. The authors should perhaps take this into consideration in their discussion.

We would like to thank the reviewer for raising this point. In addition to discussing the potential of an inhibitor targeting the UBS binding site, we also raise the possibility of exploiting the cavity above the active site as a unique target for inhibitor development (discussion page 14).

2. Figures

Figure 1: Including a schematic illustrating the structural relationship of UFC1 and UBA5, similar to the schematic depicted in the graphical abstract in Oweis et al.[6], would be helpful to the reader to visually aid the understanding of the mechanism.

We followed the reviewer's suggestion and added schematic illustration (Fig. 1B).

Figure 2: Space between the line separating the time and the variant labels needs to be adjusted as it is not consistent with the one in subfigures C and D.

The space has been adjusted as requested.

Also, a quantification of the "kinetics" of UFM1 transfer would be useful. In some cases, strong impairment of thioester formation is visible on the gel, but in other cases the difference is not readily seen. Including some type of quantification of the gel-based assays would be helpful. This also applies for the gels in Figures 4 and 5.

We have added quantifications for the gels

Additionally, a schematic diagram of UBA5 binding to UFC1 would help in understanding the model, similar to the one in Oweis et al.,[6]. Also including the linker region in this schematic would graphically depict the observations making it easier to understand.

We have added a cartoon showing the interaction of UBA5 with UFM1 and UFC1 and highlighted the linker (Fig 1B).

Figure 4 C): This figure is not readily understandable without reading the figure legend first. It could be improved by labeling it with UBA5 and perhaps including a scheme of the location UBA5 in relation to this region (e.g. zoom in on the structure to show the location of this region in regard to the entire protein).

Since the linker is flexible and not observed in our crystal structure, we have added a schematic cartoon showing the location of the sequence in the UBA5 (in the revised manuscript this is Fig 5A).

Figure 5: In the cartoon representation of the superimposition of UFC1 with seven different E2 enzymes, it would be advisable to include a legend in the figure with the color of each E2 enzyme.

We added a supplementary figure that shows superposition of each of the above seven E2s with UFC1 so the color of each E2 is clear now. (Sup. Fig. 7).

Also, providing a sequence alignment of the different E2 enzymes they used for the model (especially the colored helix region) could be included (perhaps as a supplementary figure).

We added a supplementary figure showing sequence alignment of E2s and the secondary structure elements. We have focused on helix 2 as well as on helices 3 & 4 that are missing in UFC1 but exist in all other E2s. (Sup Fig. 8).

Figure legends: Throughout the figure legends the term “SDS-PAGE” has been used, while it is obvious that the authors most likely meant “SDS-PAGE gels” or “SDS-PAGE gel analysis”. This should be corrected.

Thank you for this comment. We have modified the term to SDS-PAGE analysis

Also, the naming of the double mutant Y110 & F121A should be written as Y110A and F121 when referring to it in the text.

We have corrected this.

3. Experimental Suggestions:

To obtain the crystal structure, the authors generated a UBA5-UFC1 fusion construct. Could the authors elaborate the rationale behind this decision? Both UBA5 and UFC1 can be easily expressed in E. coli and could be added to UFM1 in the presence of ATP and MgCl₂ to generate a UFC1 thioester intermediate. Given that the thioester intermediates (Uba5-UFM1 and UFC1-UFM1) are unstable and difficult to crystallize, would it be feasible to use chemical probes[7, 8] to generate a stable thioester mimic in complex with UBA5? Furthermore, can the helix 2 of UFC1 be modified by the use of linkers (for example linkers similar to the ones reported in [9]) to support the modelling data? This approach, if structural data can be obtained, would provide further structural insights on the UFM1 thioester transfer from the E1 (UBA5) to the E2 enzyme (UFC1). Furthermore, comparison to existing literature[10] could be made.

We are happy that the reviewer raised this point in which we have invested much effort. We completely agree with the reviewer that using the fusion protein as a strategy to gain structural insight on the UBA5-UFC1 complex is not the most obvious choice. However, as shown below, we have applied several approaches, some of which are similar to the ones suggested by the reviewer, but so far with no success. Currently the fusion protein is the only strategy that worked for us. However, in the revised manuscript we have added NMR experiments that shed light on the interaction of the linker with UFC1 (Fig. 4C-E).

As the reviewer pointed out expression and purification of UBA5, UFC1 and UFM1 is not difficult and can easily yield quantities of pure proteins that are suitable for structural research. With these proteins in hand we have tried the following:

Initially, we tried to co-crystallize UFC1 and UBA5 with and without UFM1. However, since no crystals were obtained, we attempted to crystallize truncations of UBA5

together with UFC1 and UFM1, but also without success. To crystalize UFC1 with charged UBA5, we successfully purified UBA5 that is enzymatically charged with UFM1 via thioester bond (UBA5~UFM1). Interestingly, mutating C250 of UBA5 to Ser or Lys to generate an oxyester or isopeptide bond respectively did not satisfy crystallographic work. Alternatively, we used the crosslinker BMOE together with UBA5 and UFM1 G83C. We successfully purified an adduct of UBA5-BMOE-UFM1 but failed to get crystals with or without UFC1. At that point we used another strategy whereby we tried to crosslink between UBA5 and UFC1. To that end we generated functional UFC1 that has only the active site Cys (the other two cysteines were substituted to Ser). We then activated this protein with BMOE. Finally, we added UBA5 WT and obtained UBA5 linked to UFC1 via the active site Cys of each protein (as a negative control we used UBA5 C250A and showed that we did not get an adduct, highlighting the specificity to the active site Cys; see figure below).

UFC1-BMOE forms adduct with WT UBA5. UFC1-BMOE was generated by incubating UFC1 with an excess of BMOE. By using a high molar ratio of BMOE to UFC1, we guaranteed that all the UFC1 molecules were modified with BMOE on their active site Cys, thereby preventing dimerization of UFC1. Finally, UFC1-BMOE was incubated with UBA5 WT or mutant, as indicated, and adduct was probed using SDS-PAGE analysis.

Another approach we tried was based on our finding that A371 of the UBA5 linker comes near to the active site Cys of UFC1. Specifically, using A371C and the above crosslinker we obtained a complex of UBA5 linked to the active site of UFC1 via C371, but failed to get crystals.

Overall, to date only the fusion protein allowed us to gain structural insights on how the UBS binds to UBA5. However, in the revised manuscript we added an NMR experiment that provides structural insights on how the linker interacts with UFC1 (Fig. 4). The experiments show that the UBA5 linker introduces changes in the vicinity of the active site of UFC1, thereby supporting our model.

Just as a thought-provoking question: Would it be possible to generate an E2 enzyme in which the loop region (shown in Figure 3B) is deleted and then test if this E2 enzyme can function with UBA5 to charge UFM1?

This is a very interesting idea that we decided to investigate. Specifically, we designed UBCH5a that ends with helix 2, similarly to UFC1. We then tested whether the truncated UBCH5a can function with UBA5 and UFM1, but found no charging of UFM1 on UBCH5a. At that point we hypothesized that maybe the UBS on UBA5 clashes with UBCH5a since the latter, like other E2s, does not have the cavity that fits to the UBS. To that end we tested transfer of UFM1 from UBA5 (1-392) that lacks the UBS to UBCH5a, but again had no success. As noted UBCH5a similarly to other E2s does not have a cavity above the active site Cys as in UFC1 (see table 4). This cavity, which is needed for the function of UBA5 linker in UFM 1transfer, is also

missing in the truncated form of UBCH5a, thereby suggesting why transfer is not observed.

4. References:

1. Liu, G., et al., NMR and X-RAY structures of human E2-like ubiquitin-fold modifier conjugating enzyme 1 (UFC1) reveal structural and functional conservation in the metazoan UFM1-UBA5-UFC1 ubiquitination pathway. *J Struct Funct Genomics*, 2009. 10(2): p. 127-36.
2. Mizushima, T., et al., Crystal structure of Ufc1, the Ufm1-conjugating enzyme. *Biochem Biophys Res Commun*, 2007. 362(4): p. 1079-84.
3. Stewart, M.D., et al., E2 enzymes: more than just middle men. *Cell Res*, 2016. 26(4): p. 423-40.
4. Petroski, M.D., et al., Substrate modification with lysine 63-linked ubiquitin chains through the UBC13-UEV1A ubiquitin-conjugating enzyme. *J Biol Chem*, 2007. 282(41): p. 29936-45.
5. Yoo, H.M., et al., Modification of ASC1 by UFM1 is crucial for ERalpha transactivation and breast cancer development. *Mol Cell*, 2014. 56(2): p. 261-274.
6. Oweis, W., et al., Trans-Binding Mechanism of Ubiquitin-like Protein Activation Revealed by a UBA5-UFM1 Complex. *Cell Rep*, 2016. 16(12): p. 3113-3120.
7. Mulder, M.P., et al., A cascading activity-based probe sequentially targets E1-E2-E3 ubiquitin enzymes. *Nat Chem Biol*, 2016. 12(7): p. 523-30.
8. Witting, K.F., et al., Generation of the UFM1 Toolkit for Profiling UFM1-Specific Proteases and Ligases. *Angew Chem Int Ed Engl*, 2018. 57(43): p. 14164-14168.
9. Kamadurai, H.B., et al., Mechanism of ubiquitin ligation and lysine prioritization by a HECT E3. *Elife*, 2013. 2: p. e00828.
10. Page, R.C., et al., Structural insights into the conformation and oligomerization of E2~ubiquitin conjugates. *Biochemistry*, 2012. 51(20): p. 4175-87.

Reviewer #2 (Remarks to the Author):

In this manuscript, Kumar and Padala et al. focused their efforts on understanding basic mechanisms of the E1/E2/E3 enzymatic machinery responsible for protein UFMylation. The authors specifically zeroed in on an early step of the UFM conjugation cascade involving: 1) recruitment of the E2 (UFC1) to the E1 (Uba5) and 2) transfer of activated UFM1 from E1 to E2. To address these issues, the authors determined crystal structures of E2 in complex with two different E1 fragments and complemented their structural findings with a number of biochemical, biophysical, and molecular modeling studies. Together, these structures reveal the molecular basis for UBS-mediated recruitment of UFC1 to UBA5 and uncovered a surprising and unexpected role for a stretch of residues located between the UIS and UBS in promoting catalysis of transthioesterification that the authors propose functions through a desolvation mechanism. Finally, the authors used the insights gained from their collectivistudies to construct a model of the Uba5-UFC1-UFM1 complex in a catalytically active state. Overall, the manuscript is concise and the biochemical experiments in particular are elegant and carefully executed. The study is significant because it potentially fills in several key steps in our understanding of E1-E2 transthioesterification in the UFM1 system. With that said, there are some issues that should be considered and addressed to warrant publication in Nature Communications:

We would like to thank the reviewer for his/her suggestions which helped us to significantly improve our manuscript.

Key points:

The Uba5 fragment 363-377 is proposed to play a key role in transthioesterification by desolvating the E2 catalytic cysteine. However, this region is disordered in the crystal structure harboring the

347-404 fragment of Uba5. It is reasonable to assume that this could be the result of a transient interaction and the authors did perform some nice biochemistry to test their proposed role for the 363-377 fragment in desolvation of the E2 catalytic cysteine. However, given the key role of this interaction to the significance and novelty of the study I wonder if the authors could perform additional experiments that would provide deeper molecular/structural insights into the mechanism. For example, the NMR resonance assignments of UFC1 are available (BMRB Entry 6546) and it seems like it would be relatively straightforward to perform CSP experiments to map the UFC1 residues involved in the interaction with the 363-377 Uba5 fragment. A reciprocal experiment could map the Uba5 residues involved in this interaction.

We have followed the reviewer suggestion and performed CSP experiments of UFC1 with UBA5 fragments possessing the linker (see Fig. 4C-E and Sup. Fig. 9). The results support our model suggesting that the linker approaches the UFC1 active site Cys. Furthermore, the N-term helix ($\alpha 0$) of UBA5, which was earlier reported to be crucial for thermal stability of UFC1, showed perturbation in the chemical shift in the presence of UBA5 linker, suggesting that this segment may have its own importance in UFM1 transfer.

A second approach to gaining more insight to this process would be to leverage the A371C mutant the authors demonstrated was capable of cross-linking to UFC1 catalytic cysteine by attempting to crystallize a A371C mutant of the fusion protein they determined the structure of in this study. Presumably the disordered residues were present in the crystal which suggests that the crosslinked A371C species may crystallize in the same space group in a similar crystallization condition. Have the authors considered such studies

We followed the reviewer suggestion. To that end we successfully made UBA5(347-404) fused to UFC1 that has only two Cys residues at positions 371 of UBA5 and 116 of UFC1. We then induced disulfide bond between these Cys residues and confirmed intramolecular disulfide bond using SDS-PAGE. However, we did not get crystals with this fusion protein.

In the model of the Uba5-UFC1-UFM1 complex presented in Figure 5G are there any new interactions between the E1 and E2 active sites that are observed or any predicted interactions between the 363-377 Uba5 fragment and UFC1 that play a role in complex formation and catalysis? If more insight could be gained from this model that could be tested in functional studies it could increase the significance of the study significantly.

We followed the reviewer suggestions and accordingly in the revised manuscript elaborated on the ternary model (pages 12-13). As noted, the ternary model is based on our crystal structures of UFC1-UBS and UBA5 bound to UFM1. While these two structures provide critical data on how UBA5 and UFC1 binds UFM1 and UBS, respectively, they lack structural data on how the adenylation domain of UBA5 binds UFC1. To gain these missing data we docked the adenylation domain of UBA5 to UFC1 by bringing the Cys residues of the active sites one next to the other. At that point, we validated that the model satisfies our findings that the linker (i.e. UBA5 A371) can reach the active site Cys and that this model is in line with the trans-binding mechanism of UFM1 and UFC1 to UBA5. Then, by analyzing this model we found the following: 1) UFC1 interacts with the adenylation domains of both molecules of the dimeric UBA5. 2) The current conformation of UBA5 crossover loop that harbors the active site C250 has to undergo conformational changes to allow the UFC1 C116 to attack the thioester bond, but also enable Y372 to reach the cavity above the active site Cys. 3) The interaction of UFC1 with the adenylation domain of

the other UBA5 molecule is mainly mediated by the N-terminus of UBA5. 4) Although UFC1 interacts with the adenylation domains of both UBA5 molecules, these interactions are weak and possibly do not contribute to the overall interaction.

In the revised manuscript we have tested the interaction of UFC1 to UBA5 that lacks the adenylation domain. As shown in Fig. 7B, the measured K_d is similar to the one obtained with the WT UBA5, supporting the negligible contribution of the adenylation domain to UFC1 binding. Also, we measured the K_d of UFC1 to UBA5 Y372A (this mutation is in the linker and significantly reduces transfer; see Fig. 5B). As expected, we found no effect on binding to UFC1 (Fig 5D). This supports our model suggesting that the UBA5 linker is not needed for UFC1 binding but for another role, which we suggest is active site desolvation.

Minor points:

**Please show composite omit map density for the UBS in both complex structures in the supplementary material*

We have added composite omit map for both structures (Sup Fig. 2).

**Do the authors have a sense of why the UBS interacts with UFC1 in 'trans' in the AU of both fusion constructs or is this just coincidence?*

In both structures a cis mode of binding is not possible due to the constraints of the fusion protein.

The figure below shows how the UBS (red) binds UFC1 (orange). The last amino acid of UFC1 that is seen in the structure (labeled in orange sphere) is 39.5Å away from where the beginning of the UBS should be once it binds UFC1 (red sphere). Therefore, in order that the UBS that is fused to the C-terminus can bind UFC1 in cis it has to reach the position of the red sphere. However, this distance cannot be filled by the four amino acids that are not seen in the structure (the last 2 amino acids of UFC1 and two amino acids before the UBS), therefore binding in cis mode is not possible and can only be achieved in trans.

Below is the structure of the UBS (red) fused to the N-terminus of UFC1 (orange). The purple helix indicates the position of the UBS while it binds UFC1. For the UBS to bind in cis mode to UFC1, it has to move from its position in red to its position in

purple - a distance of 36.7Å. This movement is not possible since the UBS is fused to UFC.

**Many of the figures could be more clearly labeled to help readers to interpret them. Please label Fig. 1B to highlight the active site more clearly. Please also label secondary structure elements of E2*

We followed the reviewer suggestions and relabeled the figures. In Fig. 1C we added an arrow that clearly indicates the position of the active site Cys. We have added a figure showing the E2 secondary structure elements of UFC1 (Fig. 3A).

**Fig 2D- why is the molecular weight of the Uba5 347-404 fragment being added in trans so much higher than what would be expected? If there is a tag that remains?*

This is an interesting question to which we do not have an answer. However, using SEC-MALS analysis we measured the molecular weight of this fragment in solution. As shown in Sup. Fig. 6, the molecular weight of this fragment is 6620 Da, which is as expected. This suggests that the fragment behaves in solution as expected and probably its molecular weight based on the gel is an artifact.

**page 13, table 3- these are predicted pKa values, correct? If so, this should be stated.*

These are indeed predicted values and we stated this point clearly in the revised manuscript.

**please more thoroughly label figure 4B*

Done

**page 16 line 6- Fig 4A&B should be figure 5.*

We have corrected this.

**Figure 5G is confusing to me... based on Oweiss et al (Cell Reports 2016) UFMI an UFC1 binding and transthioesterification occur in trans. In this figure it appears that it is only one protomer of the Uba5 dimer that is being shown? Overall this version of the figure should be more clearly labeled. Should the UBS be red?*

We modified the figure according to the reviewer's points. In the new version of the figure (now Fig. 7A), we clearly labeled the two protomers of the dimeric UBA5 (one is in cartoon representation and the other in surface representation). In this version it is clear that UFC1 binds the UBS that arrives from one UBA5 protomer (surface representation) and that the active site of this UFC1 is approaching the active site of the other UBA5 protomer (cartoon representation).

Also the authors should consider a cartoon model for the mechanism of transthioesterification and how the new findings in this paper fit into our understanding of this process.

We followed the reviewer suggestion and made a cartoon model (Fig. 7C) that highlights our findings of how the nucleophilic activity of UFC1 is regulated by the binding of UBA5.

REVIEWERS' COMMENTS

Reviewer #1 (Remarks to the Author):

Firstly, I would like to thank the authors of the manuscript entitled “Structural basis for UFM1 transfer from UBA5 to UFC1” for elaborating on all the questions I raised. Especially, providing the experimental data underscoring their conclusions was helpful to understand the rationale for some of the experiments. Also, rewriting of some paragraphs as well as the additions I suggested have dramatically improved the readability and highlighted the importance of these findings.

While the authors have satisfactorily answered all the points, I noticed that the term “UBS” (UFC1-binding sequence) is not defined in the introduction (page 4), where it is first appears, but later in the results section (page 4). It is advisable to define this term the first time it is used.

With this minor amendment, I fully support the publication of this work given that the quality of the data as well as the novelty and impact of the authors findings is well suited for the scope of Nature Communications.

Reviewer #2 (Remarks to the Author):

In this revision Kumar et al. have significantly improved their manuscript focused on mechanisms of UFM1 transfer from Uba5 to UFC1. The figures are much improved in terms of clarity, and new analyses and discussion have been added to the manuscript that significantly increase insights gained from the study. While the authors were unable to address all of the reviewer concerns, this was largely due to technical limitations and this reviewer appreciates their strong efforts.

With that said, one of the major concerns raised during review was the lack of structural data to support the desolvation model/hypothesis proposed in this manuscript. The CSP experiments presented in the paper are an excellent addition in lieu of a structure. As a method that can provide

residue level information on amino acids involved in a protein-protein interaction (even very low affinity ones) the CSP data would ideally be used to test the biochemical data and structural modeling presented in Figures 5 and 6 that form the foundation of the author's desolvation model. Yet the authors have decided to present the CSP data prior to presentation of the functional/modeling studies and thus a thorough discussion of how the NMR data supports or refutes their data and model could not be conducted. This is an important point and is something the authors should rectify. It was satisfying to see F121 undergo chemical shifts in the presence of the Uba5 fragments, for example, but it is unclear if these are significant. Also, there is no mention of Y110 in the description of this data. It would be helpful if the residues undergoing significant chemical shifts are labeled on the surface representations shown in Figs 4 C and D and how the CSP data is consistent and inconsistent should be discussed. If CSPs for Y110 and/or F121 are not significant, why? In light of the excellent job the authors have done addressing other reviewer concerns I would be fully supportive of this manuscript being published in Nature Communications should the above analysis/discussion of the CSP data be conducted.

Reviewer 1

Firstly, I would like to thank the authors of the manuscript entitled “Structural basis for UFM1 transfer from UBA5 to UFC1” for elaborating on all the questions I raised. Especially, providing the experimental data underscoring their conclusions was helpful to understand the rationale for some of the experiments. Also, rewriting of some paragraphs as well as the additions I suggested have dramatically improved the readability and highlighted the importance of these findings.

We would like to thank the reviewer for these comments.

While the authors have satisfactorily answered all the points, I noticed that the term “UBS” (UFC1-binding sequence) is not defined in the introduction (page 4), where it is first appears, but later in the results section (page 4). It is advisable to define this term the first time it is used.

In the revised manuscript we defined the term UBS in the introduction and in the legend of Fig. 1A.

With this minor amendment, I fully support the publication of this work given that the quality of the data as well as the novelty and impact of the authors findings is well suited for the scope of Nature Communications.

We would like to thank the reviewer for this comment.

Reviewer 2

In this revision Kumar et al. have significantly improved their manuscript focused on mechanisms of UFM1 transfer from Uba5 to UFC1. The figures are much improved in terms of clarity, and new analyses and discussion have been added to the manuscript that significantly increase insights gained from the study. While the authors were unable to address all of the reviewer concerns, this was largely due to technical limitations and this reviewer appreciates their strong efforts.

With that said, one of the major concerns raised during review was the lack of structural data to support the desolvation model/hypothesis proposed in this manuscript. The CSP experiments presented in the paper are an excellent addition in lieu of a structure. As a method that can provide residue level information on amino acids involved in a protein-protein interaction (even very low affinity ones) the CSP data would ideally be used to test the biochemical data and structural modeling presented in Figures 5 and 6 that form the foundation of the author’s desolvation model. Yet the authors have decided to present the CSP data prior to presentation of the functional/modeling studies and thus a thorough discussion of how the NMR data supports or refutes their data and model could not be conducted. This is an important point and is something the authors should rectify.

In light of this comment, we have revised the order of presentation. The NMR data are now presented after the structural and biochemical data suggesting the desolvation model. In this way the reader can appreciate how the NMR data support our model.

It was satisfying to see F121 undergo chemical shifts in the presence of the Uba5 fragments, for example, but it is unclear if these are significant.

We would like to thank the reviewer for this point. In the revised manuscript we present the values that correspond to significant CSPs and how we defined them (see Fig. 6 legend and the method section). Based on these values the CSPs of

F121 are significant. Also, we have a source file with all the CSPs and the significant ones are labeled.

Also, there is no mention of Y110 in the description of this data. hereby we that prevented from us to measure CSP

Unfortunately, UFC Y110 does not have assignment thereby we cannot measure its CSPs. We included this point in the revised manuscript.

It would be helpful if the residues undergoing significant chemical shifts are labeled on the surface representations shown in Figs 4 C and D and how the CSP data is consistent and inconsistent should be discussed.

We have added to Fig. 6A a line with the secondary structure elements allowing the correlation of the CSPs of each residue to the secondary element. Also, we colored in red the bars that correspond to the CSPs of C116 and F121.

If CSPs for Y110 and/or F121 are not significant, why?

As we pointed out, the CSPs of F121 are significant with both fragments. Y110 CSPs are not measurable.

In light of the excellent job the authors have done addressing other reviewer concerns I would be fully supportive of this manuscript being published in Nature Communications should the above analysis/discussion of the CSP data be conducted.

We would like to thank the reviewer for this comment.